# Structure of bacterial phospholipid transporter MlaFEDB with substrate bound

Nicolas Coudray[1,2†], Georgia L Isom[1†], Mark R MacRae[1†], Mariyah N Saiduddin[1], Gira Bhabha[1]*, Damian C Ekiert[1,3]*

[1]Department of Cell Biology, Skirball Institute of Biomolecular Medicine, New York University School of Medicine, New York, United States; [2]Applied Bioinformatics Laboratories, New York University School of Medicine, New York, United States; [3]Department of Microbiology, New York University School of Medicine, New York, United States

**Abstract** In double-membraned bacteria, phospholipid transport across the cell envelope is critical to maintain the outer membrane barrier, which plays a key role in virulence and antibiotic resistance. An MCE transport system called Mla has been implicated in phospholipid trafficking and outer membrane integrity, and includes an ABC transporter, MlaFEDB. The transmembrane subunit, MlaE, has minimal sequence similarity to other transporters, and the structure of the entire inner-membrane MlaFEDB complex remains unknown. Here, we report the cryo-EM structure of MlaFEDB at 3.05 Å resolution, revealing distant relationships to the LPS and MacAB transporters, as well as the eukaryotic ABCA/ABCG families. A continuous transport pathway extends from the MlaE substrate-binding site, through the channel of MlaD, and into the periplasm. Unexpectedly, two phospholipids are bound to MlaFEDB, suggesting that multiple lipid substrates may be transported each cycle. Our structure provides mechanistic insight into substrate recognition and transport by MlaFEDB.

*For correspondence:
gira.bhabha@gmail.com (GB);
damian.ekiert@EKIERTLAB.ORG
(DCE)

†These authors contributed equally to this work

Competing interests: The authors declare that no competing interests exist.

## Introduction

The bacterial outer membrane (OM) is a critical barrier that protects the cell from antibiotics and other environmental threats, and protects pathogenic bacteria from the anti-microbial responses of the host. The OM is an asymmetric bilayer, with an outer leaflet of lipopolysaccharide (LPS) and a phospholipid inner leaflet. The OM is separated from the inner membrane (IM) by the periplasmic space, which contains the peptidoglycan cell wall. Although this complex envelope architecture has many advantages, it also presents many challenges for OM assembly and transport, including the need to move cargo across two lipid bilayers. Moreover, energy from ATP and the proton motive force are associated with the cytoplasm and inner membrane (IM), leaving the periplasm and OM without direct access to these conventional energy sources. Consequently, double-membraned bacteria have evolved a fascinating array of protein machines to overcome the challenge of transporting molecules beyond the IM. These include passive catalysts for OM protein insertion (BAM complex [*Knowles et al., 2009*; *Gu et al., 2016*; *Han et al., 2016*]), as well as ATP and proton-driven machines that couple energy from the cell interior to transport across the cell envelope. An elegant example of this coupling is illustrated by the LPS transport system, which couples an IM ABC (*ATP binding cassette*) transporter to a periplasmic bridge and OM complex to export newly synthesized LPS from the IM to the outer leaflet of the OM (*Okuda et al., 2012*; *Okuda et al., 2016*; *Sperandeo et al., 2017*). In contrast to the trafficking and assembly of proteins and LPS in the OM,

we know comparatively little about how phospholipids are trafficked and inserted into the inner leaflet of the OM, or how the asymmetry of the OM is maintained.

The Mla system, an ABC transporter in *E. coli* and related Gram-negative bacteria, has recently emerged as a key player in phospholipid transport across the bacterial envelope. Mla trafficks phospholipids between the IM and OM and is important for maintaining the outer membrane barrier (*Malinverni and Silhavy, 2009*; *Chong et al., 2015*; *Thong et al., 2016*; *Abellón-Ruiz et al., 2017*; *Ekiert et al., 2017*; *Isom et al., 2017*; *Shrivastava et al., 2017*; *Powers and Trent, 2018*; *Yeow et al., 2018*; *Ercan et al., 2019*; *Hughes et al., 2019*; *Kamischke et al., 2019*; *Shrivastava and Chng, 2019*). This system consists of three main parts: (1) an IM ABC transporter complex, MlaFEDB; (2) an OM complex, MlaA-OmpC/F; and (3) a soluble periplasmic protein, MlaC, which has been proposed to shuttle phospholipids between MlaFEDB and MlaA-OmpC/F (*Figure 1A*). The directionality of transport facilitated by the Mla pathway is still an area of intense research, with reports of both phospholipid import (*Malinverni and Silhavy, 2009*; *Chong et al., 2015*; *Powers and Trent, 2018*; *Yeow et al., 2018*) and export (*Hughes et al., 2019*; *Kamischke et al., 2019*). The IM complex, MlaFEDB, consists of four different proteins: MlaD, a membrane anchored protein from the MCE (Mammalian Cell Entry) protein family, which forms a homohexameric ring in the periplasm (*Thong et al., 2016*; *Ekiert et al., 2017*); MlaE (also called YrbE), a predicted integral inner membrane ABC permease; MlaF, an ABC ATPase; and MlaB, a STAS (*Sulfate Transporter and Anti-Sigma* factor antagonist) domain protein with possible regulatory function (*Kolich et al., 2020*). Crystal structures of MlaD (*Ekiert et al., 2017*) and MlaFB (*Kolich et al., 2020*) have provided insights into the function of individual domains and transporter regulation, but the structure of the transmembrane subunit, MlaE, has been lacking. MlaE lacks clear sequence similarity to proteins of known structure or function, suggesting it adopts a unique or divergent ABC transporter fold. Low-resolution cryo electron microscopy (cryo-EM) studies (*Ekiert et al., 2017*; *Kamischke et al., 2019*) have established the overall shape of the complex, but have not shed much light on how the various subunits of the MlaFEDB complex assemble and function. Thus, a structure of the MlaFEDB complex may provide important insights into the mechanisms of bacterial lipid transport, as well as the evolution and function of the MlaE/YrbE transporters, which are conserved from double-membraned bacteria to chloroplasts.

## Results

### Overview of the MlaFEDB structure

To address how Mla drives lipid transport, we overexpressed the *mla* operon (*Figure 1B*), reconstituted the MlaFEDB ABC transporter complex in lipid nanodiscs containing *E. coli* polar lipids (see Materials and methods), and determined the structure using single-particle cryo-EM, with a nominal resolution of 3.05 Å (*Figure 1C–E*, *Figure 1—figure supplements 1* and *2*, *Supplementary file 1*). Although MlaFEDB was expected to exhibit twofold symmetry, initial reconstructions showed clear asymmetry in our maps, which we then refined without applying symmetry (*Figure 1D*, *Figure 1—figure supplements 1* and *2*, *Supplementary file 2*). Local resolution analysis showed that the entire complex is well-defined at ~2.8–3.5 Å resolution (*Figure 1D*, *Figure 1—figure supplement 3A*), allowing us to build a nearly complete model for MlaFEDB (*Figure 1F*), including a high-resolution structure of the MlaE transmembrane subunit. In addition, we resolved both coils of the membrane scaffold protein (MSP) belt surrounding the nanodisc (*Bayburt et al., 2002*) using the map filtered at 6 Å, thereby clearly defining the position of the transmembrane domain (*Figure 1E*, *Figure 1—figure supplement 3A,B*).

The MlaFEDB transporter is significantly larger and more complicated than most other structurally characterized ABC transporters, consisting of a total of 12 polypeptide chains from four different genes, with a stoichiometry of $MlaF_2E_2D_6B_2$. At the center of the complex, a core ABC transporter module is formed from two copies each of the MlaF ATPase and the MlaE transmembrane domains (TMDs). Outside this ABC transporter core, MlaFEDB contains additional subunits not found in other ABC transporters: MlaD on the periplasmic side of the IM, and MlaB in the cytoplasm. A homohexameric ring of MCE domains from MlaD sits atop the periplasmic end of MlaE like a crown. This MCE ring is anchored in place by six MlaD transmembrane helices, which dock around the periphery of the MlaE TMDs (*Figure 1F*). On the cytoplasmic side, each of the MlaF ATPase subunits is bound to

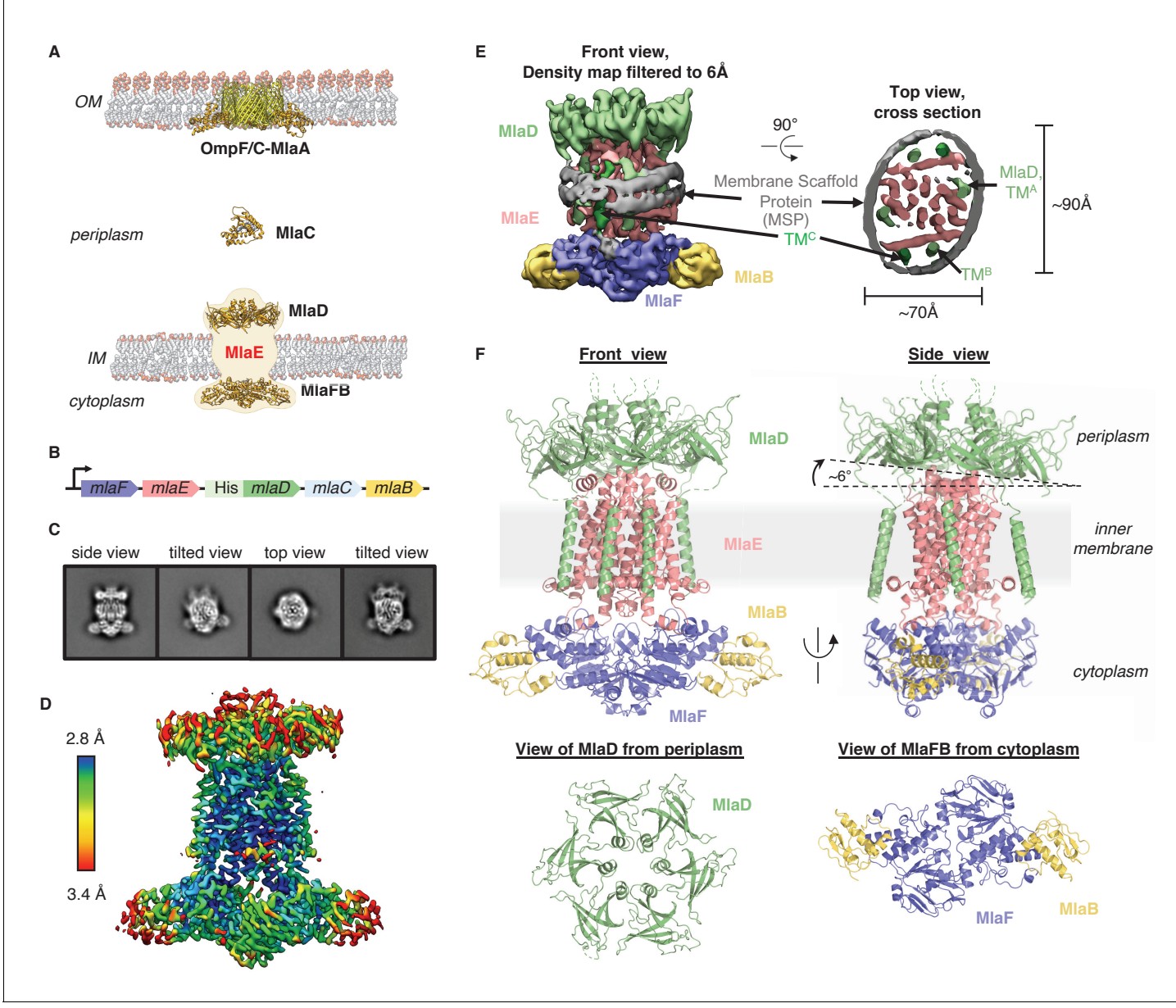

**Figure 1.** Cryo-EM structure of the MlaFEDB complex. (**A**) Schematic of the Mla pathway (adapted from *Kolich et al., 2020*). The OmpC/F-MlaA complex (PDB 5NUP), periplasmic shuttle protein MlaC (PDB 5UWA), and MlaFEDB complex (PDB 5UW2 and 6XGY, EMDB-8610) are shown. (**B**) Schematic of the MlaFEDCB operon with N-terminal His-tag on MlaD, as reported previously (*Ekiert et al., 2017*). (**C**) 2D class averages from single-particle cryo-EM analysis of MlaFEDB in nanodiscs. (**D**) Final EM density map of MlaFEDB, colored by local resolution (EMD-22116). (**E**) Density map of MlaFEDB filtered to 6 Å, showing membrane scaffold protein (MSP) belts surrounding the edge of the lipid nanodisc. MlaF, slate blue; MlaE, pink; MlaD, green; MlaB, yellow; MSP, gray. The packing of the six TM helices from the MlaD subunits around the periphery of the MlaE core is apparent in the Top View. (**F**) Overview of the MlaF$_2$E$_2$D$_6$B$_2$ model (PDB 6XBD); colors as in **E**. Regions of disorder in MlaD linkers and C-termini are indicated by green dashed lines. The MlaD ring is tilted relative to MlaFEB, resulting in a complex that is asymmetric overall.

The online version of this article includes the following figure supplement(s) for figure 1:

**Figure supplement 1.** Cryo-EM data processing workflow.

**Figure supplement 2.** Cryo-EM data and analysis.

**Figure supplement 3.** Representative density for MlaFEDB complex.

**Figure supplement 4.** Comparison of MlaFB from X-ray and Cryo-EM structures.

**Figure supplement 5.** Extra densities in the transmembrane region.

MlaB, a STAS domain protein that was recently reported to act as regulator of the MlaFEDB transporter (*Kolich et al., 2020*). The overall structure of the MlaF$_2$B$_2$ module is very similar to our recent MlaFB X-ray structure (PDB: 6XGY), apart from a small relative rotation between the MlaF helical and catalytic subdomains (*Figure 1—figure supplement 4A*). This rotation is similar to motions described in other ABC transporters (*Karpowich et al., 2001*; *Smith et al., 2002*; *Orelle et al., 2010*). An unusual C-terminal extension of each MlaF protomer wraps around the neighboring MlaF subunit and docks near the MlaFB interface, almost identical to the domain-swapped 'handshake' motif observed in the crystal structure of the MlaF$_2$B$_2$ subcomplex (*Figure 1—figure supplement 4B*; *Kolich et al., 2020*). While the MlaFEB subcomplex exhibits near-perfect twofold rotational symmetry at this resolution, the MlaD ring is clearly tilted relative to MlaE, resulting in a misalignment of the twofold symmetry axis of MlaFEB and the pseudo-sixfold axis of MlaD by approximately 6° (*Figure 1F*).

## MlaE is distantly related to the TMDs of other ABC transporters

The transmembrane subunits of ABC transporters play a central role in determining the transport mechanism and substrate specificity. Consequently, the structure of the MlaE subunit is of particular interest. Our cryo-EM structure reveals that the core TMD of MlaE consists of five transmembrane helices (TM1 - TM5) (*Figure 2A and E–G*). A coupling helix (CH) in the cytoplasm connects TM2 and TM3 and mediates the interaction between the TMDs of MlaE and the MlaF ATPase subunits (*Figure 1—figure supplement 4C*). A small periplasmic helix (PH) is found between TM3 and TM4 at the periplasmic side of MlaE. Two additional N-terminal helices are membrane embedded, which we call interfacial helices 1 and 2 (IF1 and IF2 [*Chen et al., 2020*]; discussed in more detail below). The IF1 helix is a 30-residue long, amphipathic N-terminal helix that lies parallel to the membrane within the cytoplasmic leaflet, and extends along the width of the MlaE dimer (*Figures 2A* and *3B*). IF2 is angled relative to the plane of the membrane and is separated from TM1 by a kink within the lipid bilayer. Although the C-terminal portion of IF1 interacts with TM3 and TM4, the N-terminal half projects outward into the surrounding membrane, creating a cleft between the core TMD and the IF1 helix (*Figure 3A,B*). Additional EM densities were observed in this cleft (*Figure 1—figure supplement 5*), which may be phospholipids or other molecules; these ligands were not modeled explicitly as their identities are ambiguous.

Despite negligible sequence similarity, the core MlaE fold is related to the TMDs of several other ABC transporters. MlaE most closely resembles the LPS exporter LptF/LptG (*Thomas et al., 2020*) and the macrolide antibiotic efflux pump MacB (*Crow et al., 2017*; *Fitzpatrick et al., 2017*; *Okada et al., 2017*; *Figure 2B–J*), but also shares similarities with the glycolipid flippases Wzm (*Bi et al., 2018*; *Caffalette et al., 2019*) and TarG (*Chen et al., 2020*) and the eukaryotic ABCA/ABCG families (*Figure 2—figure supplement 1*). However, MlaE also displays notable differences. First, previously determined structures from the above families contained either four TM helices (MacB) or six TM helices (all the rest). In contrast, MlaE is intermediate between these two groups; MacB contains TM1-TM4, MlaE contains TM1-TM5, and other transporters contain TM1-TM6 (*Figure 2*, *Figure 2—figure supplement 1*). Second, in both MlaE and LptFG, TM5 exists as one continuous helix, whereas TarG, Wzm, and ABCA/ABCG have a small insertion near the periplasmic end that results in a pair of reentrant helices (*Figure 2—figure supplement 2A*). Third, MlaE has a membrane-embedded helix preceding TM1, which we call IF2 (also called CnH [*Lee et al., 2016*], IF [*Bi et al., 2018*], or 'elbow helix'). In MacB, TarG, Wzm, and ABCA/ABCG transporters, IF2 forms an amphipathic helix that runs roughly parallel to the membrane within the cytoplasmic leaflet, followed by a sharp turn and a separate TM1, which spans nearly the entire bilayer (*Figure 2—figure supplement 2B*). In contrast, LptFG has no clear IF2 counterpart, as these TMDs begin with TM1 forming a long, continuous helix (*Figure 2—figure supplement 2B*). In MlaE, this region adopts an intermediate configuration, where IF2 and TM1 are both involved in the first traverse of the membrane, but these two segments are distinguished by a clear kink in the middle of the bilayer (*Figure 2—figure supplement 2B*). Overall, the structure of the transmembrane region of MlaE differs significantly from previously determined structures, but has at its core a transporter domain conserved across a structurally diverse group of ABC transporters and shared between bacteria and eukaryotes.

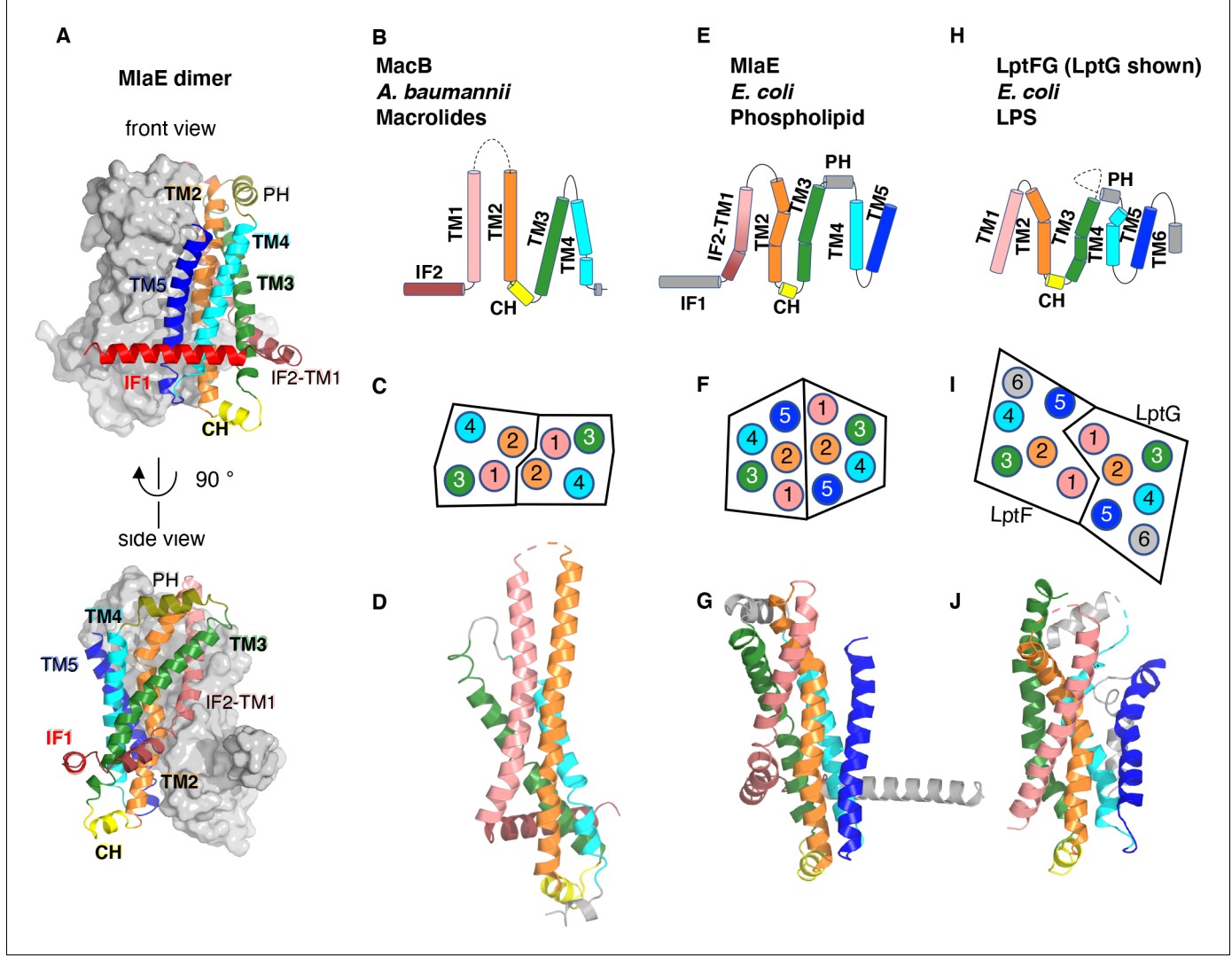

**Figure 2.** Topology and fold of MlaE. (A) MlaE dimer, with one protomer represented as surface, and the other as cartoon. (B–J) comparison of MlaE (PDB 6XBD) with the related transmembrane domains of ABC transporters, MacB and LptFG (PDB 5GKO and 6MHZ). (B, E, H) Topology diagrams. CH, coupling helix; PH, periplasmic helix; IF, interfacial helix (also called connecting helix in ABCG transporters); TM, transmembrane helix. (C, F, I) Schematics representing helices at the dimer interface, viewed from the periplasm (each circle represents a helix). (D, G, J) Cartoon view of monomer. See *Figure 2—figure supplement 1* for comparisons with additional related transporters.

The online version of this article includes the following figure supplement(s) for figure 2:

**Figure supplement 1.** Comparison of MlaE to other ABC transporters.

**Figure supplement 2.** Structural variation in MlaE TMD compared to other ABC transporters.

## Interactions between MlaE and MlaD

MlaE has three main modes of interaction with MlaD (*Figure 3A*). First, the MlaE core TMDs interact with two of the MlaD TM helices; second, the IF1 helices of the MlaE dimer interact with the remaining four MlaD TM helices; and third, the periplasmic end of the MlaE dimer interacts with the MlaD ring. Due to the symmetry mismatch between the pseudo-sixfold symmetric MlaD hexamer and the two-fold symmetric MlaFEB module, the six identical transmembrane helices of MlaD interact with MlaE in three non-equivalent ways (*Figure 3B*). The MlaD TMs from chains A and D (MlaD-TM^A/D) are closely packed against the MlaE TMDs on opposite sides of the complex. The remaining four TM helices from MlaD are largely isolated in the membrane, and their main interactions are with the MlaE IF1 helices via helix crossing interactions, with MlaD-TM^B/E and MlaD-TM^C/F contacting IF1 residues 6–14 and 17–25, respectively (~84° crossing angle, *Figure 3A*, *Figure 3—figure supplement*

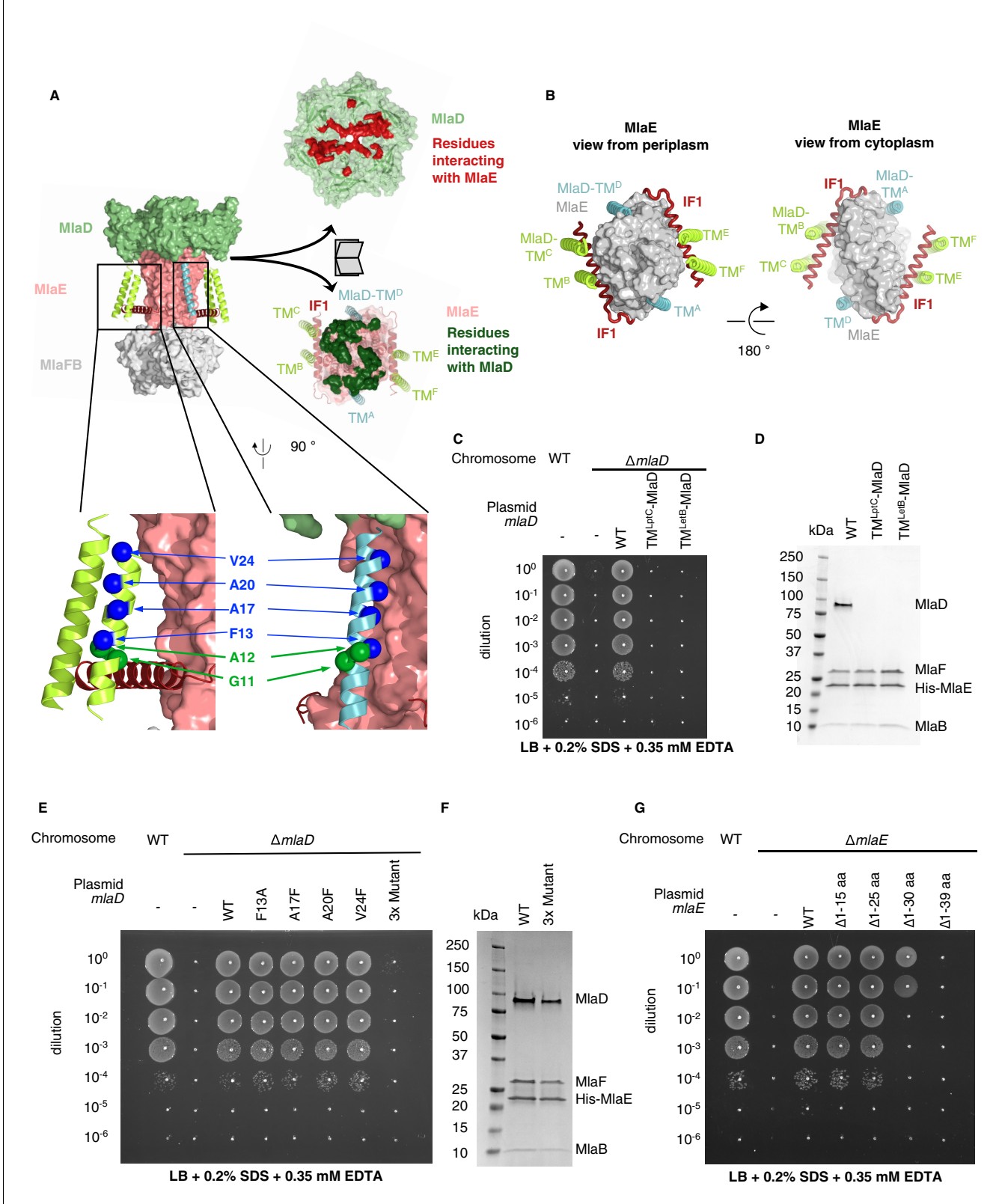

**Figure 3.** TM helices of MlaD are important for interaction with MlaE. (**A**) Structure of MlaFEDB complex, highlighting interacting regions between MlaE and MlaD. MlaD-TM^(A/D) (cyan helices) interact with the core domain of MlaE (pink surface); MlaD-TM^(B/C/E/F) (green helices) interact mostly with IF1 (red helices). The periplasmic end of the MlaE dimer interacts with the MlaD ring (shown in 'open book' representation, right). MlaD residues that interact with MlaE are shown in red, and MlaE residues that interact with MlaD are shown in green, as determined using COCOMAPS (***Vangone et al.,***

*Figure 3 continued on next page*

*Figure 3 continued*

*2011*). Below, a zoom of the positions on the TM helices of MlaD mutated to test interactions either with IF1 (green spheres) or the TM helices of MlaE (blue spheres) (related to panel E), (F) and (*Figure 3—figure supplement 2C-E*). (B) Periplasmic and cytoplasmic views highlighting the interaction between helices of MlaE and MlaD. Helices colored as in (A). (C) Genetic complementation of *mlaD* knockout with MlaD TM chimeras. 10-fold serial dilutions of the indicated cultures spotted on LB plates containing SDS+EDTA at the concentrations indicated, and incubated overnight. The *mlaD* knockout does not grow in the presence of SDS+EDTA, but can be rescued by the expression of WT MlaD from a plasmid. MlaD chimeras containing LptC or LetB TM helices fail to complement. Corresponding controls plated on LB only can be found in *Figure 3—figure supplement 2A*. (D) SDS-PAGE of recombinantly expressed and purified complexes formed in the presence of WT MlaD or MlaD chimeras containing LptC or LetB TM helices. (E) Genetic complementation of *mlaD* knockout with MlaD TM mutants. The assay is performed as in (C). F13A, A17F, A20F, V24F mutants complement, while a triple mutant (A17F/A20F/V24F; '3x Mutant') fails to complement. Corresponding controls plated on LB only can be found in *Figure 3—figure supplement 2C*. (F) SDS-PAGE of recombinantly expressed and purified complexes formed in the presence of WT MlaD or the MlaD 3x Mutant. (G) Genetic complementation of *mlaE* knockout with MlaE IF1 truncation mutants. The assay is performed as in (C). Small deletions of the N-terminus of MlaE IF1 are tolerated, while larger deletions impair or completely abolish growth. Corresponding controls plated on LB only can be found in *Figure 3—figure supplement 2D*.

The online version of this article includes the following figure supplement(s) for figure 3:

**Figure supplement 1.** Sequence conservation in MlaD and MlaE.
**Figure supplement 2.** Extended data for MlaD mutants.
**Figure supplement 3.** Extended data for MlaE mutants.

*1*). The residues of IF1 interacting with the MlaD-TM$^{B/E}$ and MlaD-TM$^{C/F}$ helices form two well-conserved LxxF/LG motifs, with the conserved Gly11 of each MlaD TM making close packing interactions with the two conserved Gly residues from IF1 (Gly10 and Gly21).

To test whether the MlaD TM helices are important for MlaFEDB complex assembly, we generated chimeras in which we replaced the native TM helix of MlaD with a TM helix expected to make no direct interactions with MlaE [from the *E. coli* IM proteins LptC (TM$^{LptC}$) or LetB (TM$^{LetB}$)]. We tested the ability of these MlaD chimeras to complement an *mlaD* knockout strain of *E. coli*. Mutations in components of the Mla pathway exhibit a substantial growth defect in LB medium in the presence of SDS+EDTA (*Malinverni and Silhavy, 2009*), which can be complemented with WT *mlaFEDCB* on a plasmid. We found that neither of the MlaD chimeras were able to restore growth of an *mlaD* knockout strain in the presence of SDS+EDTA (*Figure 3C*, *Figure 3—figure supplement 2A*). To assess whether these MlaD chimeras are still capable of interacting to form MlaFEDB complexes, we recombinantly overexpressed the *mlaFEDCB* operon encoding either WT or chimeric MlaD, and purified the resulting complexes using an affinity tag on MlaE. SDS-PAGE showed that MlaE co-purified with MlaB, MlaF and WT MlaD, but MlaD chimeras did not co-purify (*Figure 3D*), despite robust expression of hexameric MlaD in the membrane fraction (*Figure 3—figure supplement 2B*). Thus, the mere presence of MlaD hexamers anchored to the membrane is not sufficient to complement an *mlaD* knockout, but rather MlaD appears to require its native TM helix in order to assemble and function in complex with MlaFEB. These results suggest that the MlaD TM helix interactions with MlaE drive specificity in the formation of the complex.

While MlaD-TM$^{A/D}$ helices interact intimately with the core MlaE TMD, MlaD-TM$^{B/E}$ and MlaD-TM$^{C/F}$ interact more loosely with MlaE, mostly through IF1. We constructed a series of mutations in MlaD and MlaE to probe the relative contributions of these interactions in the TM region. Single mutations in MlaD predicted to interfere with the interactions between MlaE and MlaD-TM$^{A/D}$ by creating steric clashes (Ala17Phe, Ala20Phe, or Val24Phe) or removing a bulky hydrophobic residue (Phe13Ala) had no effect on the complementation of an *mlaD* knockout strain (*Figure 3A,E* and *Figure 3—figure supplement 2C*). In contrast, a triple mutant (Ala17Phe/Ala20Phe/Val24Phe) failed to complement, but yielded intact MlaFEDB complexes when overexpressed and purified (*Figure 3E and F*). These results indicate that the interactions made by the MlaD-TM$^{A/D}$ helices may be redundant for MlaFEDB complex assembly because the remaining MlaD-TM$^{B/E}$ and MlaD-TM$^{C/F}$ helices can compensate. However, the interactions made by the MlaD-TM$^{A/D}$ helices are required for transporter function in cells, suggesting a possible role in mechanism, such as modulating conformational change. Next, we assessed the role of the interactions between the IF1 helix of MlaE and MlaD-TM$^{B/E}$ and MlaD-TM$^{C/F}$. We generated truncation mutants of MlaE IF1 with 15 or 25 residues deleted from the N-terminus of IF1 (Δ1–15 aa and Δ1–25 aa), thereby removing the MlaD-TM$^{C/F}$-binding site or both the MlaD-TM$^{C/F}$ and MlaD-TM$^{B/E}$ binding sites, respectively (*Figure 3—figure supplement*

*3A*). We used a similar genetic complementation assay to the one described above, but for an *mlaE* knockout, to assess the function of these variants in cells. We observed that both the Δ1–15 aa and Δ1–25 aa mutants fully restored growth of an *mlaE* knockout under these conditions, similar to complementation by the WT operon (*Figure 3G*, *Figure 3—figure supplement 3B*). Consistent with these results, when point mutations were introduced into MlaD at positions predicted to disrupt interactions with both conserved LxxF/LG motifs in IF1, all mutants tested were able to fully complement an *mlaD* knockout (*Figure 3A*, *Figure 3—figure supplement 2D,E*). All the MlaE truncation mutants expressed well, and formed complexes with MlaD, MlaF, and MlaB (*Figure 3—figure supplement 3C*), although we noted that all the mutants appeared to incorporate less MlaFB into the complex. The significance of MlaFB destabilization is not clear, although binding of MlaFB to MlaE was previously proposed to be weaker than is typically observed for ABC transporters, and reversible association/dissociation of the complex may be a mechanism of MlaFEDB regulation (*Kolich et al., 2020*). Thus, while the interactions between the IF1 helix of MlaE and the MlaD-TM$^{B/E}$ / MlaD-TM$^{C/F}$ helices are not strictly required, they likely still contribute to MlaFEDB complex stability. Taken together, our data suggest that the TM regions of MlaD cooperate to facilitate the assembly and stability of the MlaFEDB complex, and that the interactions formed by TM$^{A/D}$ may have an additional role in the function of the MlaFEDB transporter that remains to be unraveled. Finally, while the region of IF1 that interacts with MlaD appears to be dispensable, MlaE mutants with larger deletions only partially restored growth (Δ1–30 aa), or failed to complement (Δ1–39 aa), suggesting that the C-terminus of IF1 and/or the following loop has an essential function (*Figure 3G*).

## MlaE adopts an outward-open conformation

While the vast majority of ABC transporter structures adopt an inward-open conformation in the absence of nucleotide (*Gerber et al., 2008*; *Kadaba et al., 2008*; *Aller et al., 2009*; *Manolaridis et al., 2018*), our structure of MlaE is in the outward-open state (*Figure 4A*). This uncommon configuration was previously observed in the related transporters LptFG (*Li et al., 2019*; *Owens et al., 2019*) and MacB (*Fitzpatrick et al., 2017*), suggesting that Mla, Lpt, and MacAB may share some mechanistic features. The narrow outward-open pocket within MlaE encloses a volume of ~750 Å$^3$ (estimated using CASTp [*Tian et al., 2018*]) and is primarily formed by TM1 and TM2, with some contribution from TM5 (*Figure 4A*). This is similar to LptFG, where TM1, TM2, and TM5 form a much larger pocket (volume of ~3000 Å$^3$, estimated using CASTp [*Tian et al., 2018*], PDB: 6MHU [*Li et al., 2019*]) at the periplasmic side of the complex for LPS binding. The pockets in both transporters are largely hydrophobic in nature, consistent with binding to lipid substrates, although in LptFG the rim has a pronounced positive charge proposed to interact with phosphates on the LPS inner core (*Li et al., 2019*), while MlaE is more neutral.

MlaD forms a hexameric MCE ring with a hydrophobic pore at the center (*Ekiert et al., 2017*). Similar hydrophobic tunnels have been observed through the MCE rings of PqiB and LetB (*Ekiert et al., 2017*; *Isom et al., 2020*), and phospholipids have been cross-linked inside the tunnel of LetB. Indeed, in our structure of the MlaFEDB complex, the pore of MlaD and the outward-open pocket of MlaE line up with each other, resulting in a continuous hydrophobic pathway that runs from the pocket of MlaE, through MlaD, and out into the periplasm (*Figure 4B*). Structurally diverse lipid transport domains have also been resolved for several other ABC transporters (*Qian et al., 2017*; *Li et al., 2019*; *Owens et al., 2019*; *Figure 4—figure supplement 1*), and are proposed to facilitate the movement of hydrophobic molecules through the aqueous extracellular/periplasmic environment.

## Lipids are bound in the substrate-translocation pathway

Additional density was apparent in the outward-open pocket of MlaE, which is the right size and shape to accommodate two diacyl phospholipids (*Figure 4C,D*). We modeled these as the most abundant PL species in *E. coli*, phosphatidylethanolamine, and these lipids make extensive contacts with both MlaE subunits, as well as MlaD (*Figure 4—figure supplement 2A*; see Materials and methods for additional details). Unlike recent structures of the LPS exporter bound to LPS, where all of the acyl chains project downward into the hydrophobic pocket of LptFG (*Li et al., 2019*; *Tang et al., 2019*), the lipids bound to MlaFEDB appear to be trapped in different conformations (*Figure 4C*), perhaps intermediates in the process of being transferred between MlaE and the

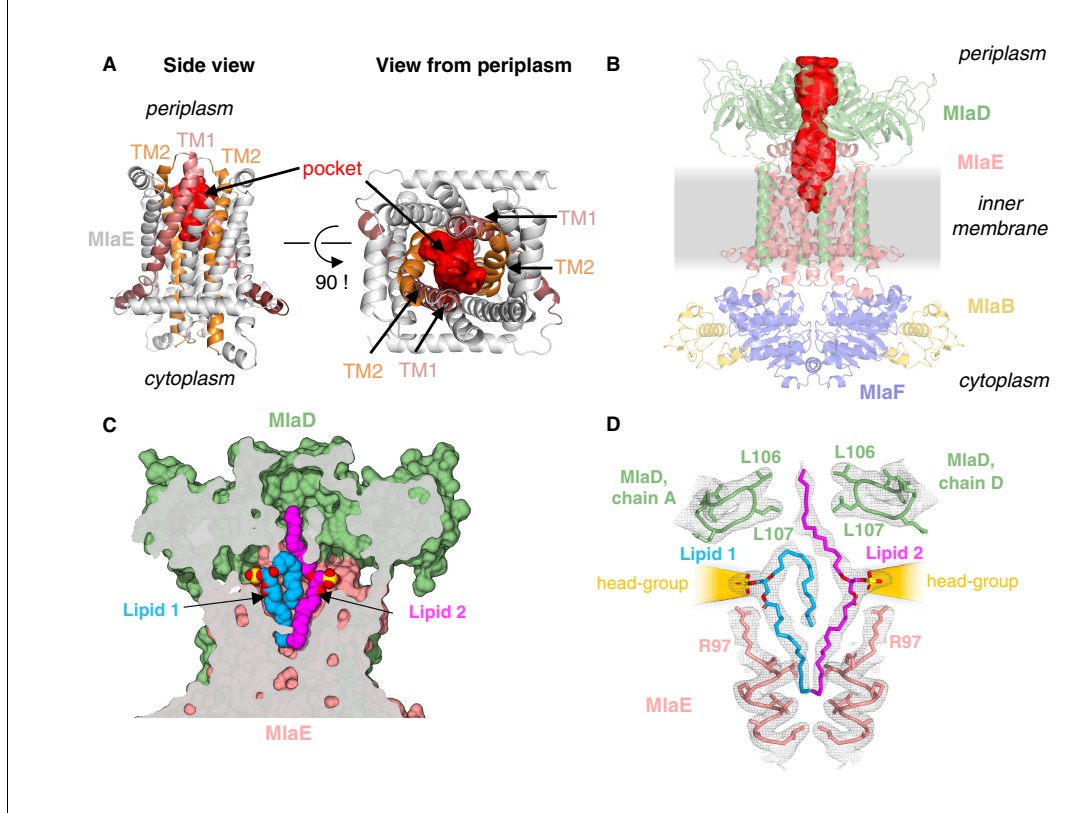

**Figure 4.** Lipids are bound in an outward-open pocket formed by MlaE and MlaD. (**A**) Side view (left) and view from periplasm (right) of MlaE dimer highlighting the outward-open pocket formed by TM1 (salmon helices) and TM2 (orange helices). The boundary of the substrate-binding pocket was estimated using CASTp (*Tian et al., 2018*), and is displayed as a red surface. (**B**) Side view of MlaFEDB complex, showing the continuous hydrophobic channel running from the substrate-binding pocket in MlaE to the periplasmic space, through the pore of MlaD (red, tunnel boundary estimated with HOLLOW [*Ho and Gruswitz, 2008*]). (**C**) Side view cross-section of the hydrophobic channel between MlaE and MlaD showing two bound phospholipids in blue and magenta. (**D**) Lipids modeled with surrounding structural elements and key residues highlighted. EM density map is shown as mesh, at the same contour level for lipids and surrounding regions. The lateral channels that could accommodate lipid head groups are indicated by orange cones.

The online version of this article includes the following figure supplement(s) for figure 4:

**Figure supplement 1.** Structural diversity of extracytoplasmic lipid transport domains.

**Figure supplement 2.** Asymmetry of lipid interactions and asymmetry of MlaD in the MlaFEDB complex.

**Figure supplement 3.** Point mutations in MlaE.

**Figure supplement 4.** MlaC structures in which either one or two lipids are bound.

**Figure supplement 5.** Comparison of lipid modeling into MlaFEDB map EMDB-30355 (*Chi et al., 2020*).

MlaD pore. In one lipid molecule (lipid 1), both acyl chains are bound in the pocket of MlaE (*Figure 4C,D*). Strikingly, the second lipid (lipid 2) adopts an extended conformation, with one acyl chain reaching down into the MlaE pocket, while the other projects upwards to insert into a constriction in the MlaD pore formed by Leu106 and Leu107 (*Figure 4D*), which have previously been implicated in MlaD function (*Ekiert et al., 2017*). The upward-facing acyl chain of lipid 2 is sandwiched between two tyrosine residues (Tyr81 from each MlaE subunit), and these residues also contact one of the acyl chains of lipid 1. Mutations of Tyr81 to either a smaller (Ala) or a larger (Trp) hydrophobic residue had no effect on *E. coli* growth in the presence of SDS+EDTA (*Figure 4—figure supplement 3*), although a Tyr81Glu mutation was recently reported to impair MlaFEDB function (*Tang, 2020*) (see Discussion).

In contrast to the hydrophobic fatty acid tails, which are completely buried within the MlaE-MlaD tunnel, the polar head group of each lipid projects outwards through lateral solvent accessible channels on opposite sides of the complex. Beyond the phosphoglycerol core, the density for the lipid head groups in the lateral channels of MlaFEDB are not well resolved, and the only noteworthy

interaction is a salt bridge formed between Arg97 of MlaE and the head group phosphate (*Figure 4D*). Weaker density beyond the phosphate may reflect heterogeneity in the lipid species bound to MlaFEDB in our structure, and/or that the interactions between the headgroup and nearby MlaE and MlaD residues may be weak and non-specific. The crystal structure of MlaC bound to phospholipid revealed a similar binding mode and lack of head group specificity (*Ekiert et al., 2017*). Thus, rather than mediate the binding of specific lipids, the lateral channels may instead serve as a non-specific cavity to accommodate a range of polar head groups during the 'lipid gymnastics' (*Neumann et al., 2017*) that may need to occur to translocate lipids between the MlaE pocket and the MlaD pore, which likely involve flipping the lipids upside down (see Discussion). In addition to interactions with the lipid head groups, Arg97 is part of a cluster of conserved charged residues, including Glu98, Lys205, and Asp250, which form salt bridges buried in the hydrophobic core of MlaE and are part of a larger polar interaction network including Gln73, Asp198, and Thr254 (*Figure 3—figure supplement 1C*, *Figure 4—figure supplement 3A*). To probe the potential role of these residues in MlaFEDB function, we mutated Arg97, Glu98, Lys205, and Asp250 individually to alanine, and found that these mutations had no effect on *E. coli* growth in the presence of SDS +EDTA (*Figure 4—figure supplement 3*; see Discussion). Thus, the role of these conserved residues remains unclear.

The presence of two lipid densities in our structure raises the possibility that the Mla system may transport two substrates per transport cycle. Structures of the periplasmic lipid carrier protein, MlaC, have been determined with either one or two diacyl phospholipids bound (*Ekiert et al., 2017*; *Figure 4—figure supplement 4*), and a structure of apo *E. coli* MlaC revealed a clamshell-like motion resulting in significant changes in the volume of the lipid binding pocket (*Hughes et al., 2019*). The different architectures and conformational states of the MlaC pocket suggest that it may accommodate one or two phospholipids, or larger lipid molecules. Indeed, previous studies have suggested that cardiolipin may be a substrate of the Mla system (*Kamischke et al., 2019*), and cardiolipin is detected by TLC in lipid extracts from components of the Mla system (*Hughes et al., 2019*). Together with previous functional data, the presence of two phospholipids/four acyl chains bound in our MlaFEDB structure raises the possibility that the Mla system may also be capable of transporting tetra-acyl lipids, such as cardiolipin. We note, however, that cardiolipin bound to MlaE would have to adopt a somewhat different configuration from the lipids observed in our structure, as the two head group phosphates in our structure are ~20 Å apart, while they would be expected to be ~6–7 Å apart in cardiolipin.

Viewed from the side, the MlaD ring is tilted with respect to MlaFEB by approximately 6° (*Figure 1F*) and deviates from the expected sixfold symmetry observed in the crystal structure of MlaD and other MCE proteins (*Figure 4—figure supplement 2B,C*). Differences between subunits that result in symmetry breaking include re-organization of two of the six pore lining loops (containing Leu107) as well as domain level rearrangements in the ring (*Figure 4—figure supplement 2B,C*). This is particularly surprising, as the MlaFEB module is twofold symmetric in our EM structure, and the crystal structure of the MlaD ring in isolation exhibited near perfect sixfold symmetry. This raises the question: what is breaking the symmetry in our MlaFEDB structure? The clear asymmetric density for the two bound phospholipids suggests that the asymmetry of the MlaD ring and the configuration of the lipids is correlated; otherwise, the cryo-EM reconstruction would yield twofold symmetric lipid densities to match the twofold symmetric features of the MlaFEB module. Thus, the asymmetry in MlaD appears to arise from its asymmetric interactions with lipid 1 at the interface of MlaE and MlaD. Leu107 from MlaD chain F makes hydrophobic interactions with one of the fatty acid tails, perhaps stabilizing this side of the MlaD ring in closer proximity to MlaE and the lipid-binding pocket. The resulting conformational changes in the MlaD ring could be important for lipid translocation through the channel or perhaps even modulating the binding of MlaC to the transporter and facilitating lipid transfer between MlaD and MlaC (see Discussion).

To assess whether phospholipids are bound in the pocket of MlaE in cells, we utilized a site-specific photocross-linking method (*Isom et al., 2020*) to detect binding of radiolabeled phospholipids in vivo. We incorporated the unnatural photocross-linking amino acid *p*-benzoyl-L-phenylalanine (BPA) (*Chin et al., 2002*) into the MlaE protein at five positions in the lipid-binding site (Leu70, Val77, Leu78, Tyr81, and Leu99), as well as Phe209 (protected in the MlaE core; not expected to contact lipids) or Trp149 (membrane exposed; expected to contact bulk membrane lipids) (*Figure 5A*). After cross-linking in cells that were grown in the presence of $^{32}$P orthophosphate to

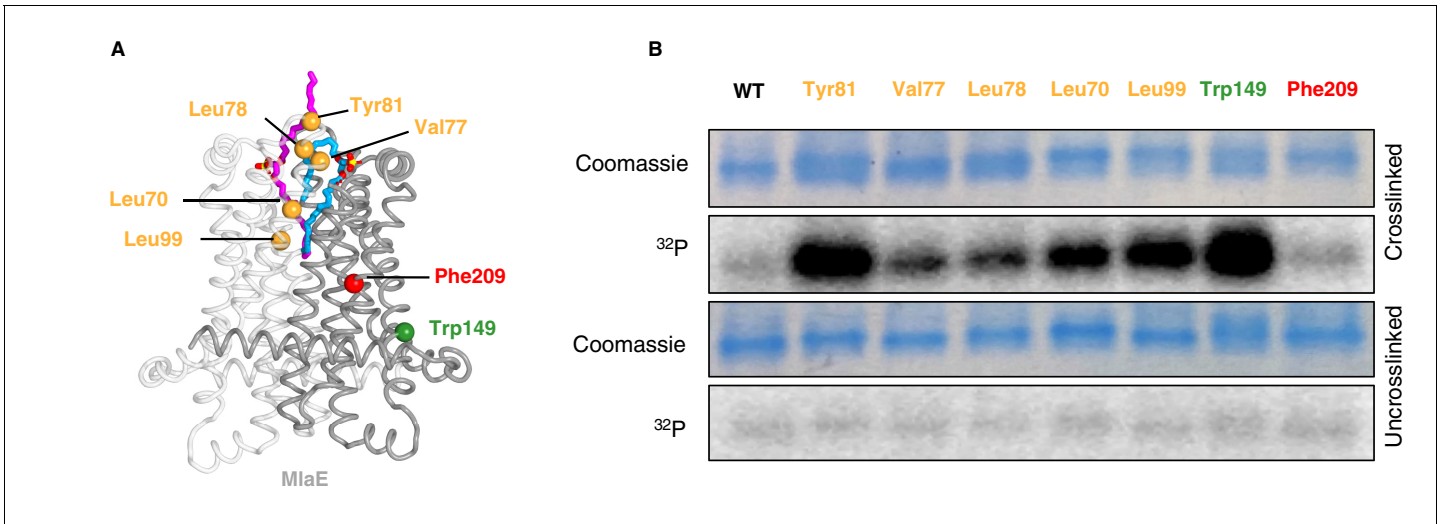

**Figure 5.** In vivo photocross-linking of substrates in MlaFEDB. (A) MlaE dimer (gray cartoon) showing sites of BPA cross-linker incorporation (spheres). Orange, residues in the lipid-binding pocket; red, residue buried within the MlaE core, designed as a negative control; green, residue facing the membrane environment, designed as a positive control. Bound lipids are shown in magenta and blue sticks. (B) SDS-PAGE of purified WT MlaFEDB and BPA mutants, either cross-linked or uncross-linked, and visualized by Coomassie staining (protein) or phosphorimaging ($^{32}$P signal). Band corresponding to the MlaE monomer is shown here, and full gels are shown in *Figure 5—figure supplement 1*.

The online version of this article includes the following figure supplement(s) for figure 5:

**Figure supplement 1.** Uncropped gels of the in vivo photocross-linking assay shown in *Figure 5*.

**Figure supplement 2.** Western blot against MlaE, showing positions of MlaE monomer and cross-linked MlaE dimer.

label total phospholipids, these MlaFEDB complexes were purified and analyzed by SDS-PAGE. We observed both a monomeric and dimeric form of MlaE, the latter of which was enriched in cross-linked samples where BPA had been incorporated at the dimer interface (*Figure 5—figure supplement 1*, *Figure 5—figure supplement 2*). However, as the level of dimerization was variable between mutants, we focused only on the monomeric band for our analysis (*Figure 5B*). Cross-linking at Trp149 and Phe209 resulted in high and low $^{32}$P signal, respectively, indicative of an abundance of phospholipids near the membrane-exposed Trp149 and very few phospholipids near the buried Phe209, as expected (*Figure 5B*). $^{32}$P incorporation into MlaE was induced by cross-linkers at all five positions in the outward-open lipid binding pocket, with particularly high signals for Tyr81 and Leu99 (*Figure 5B*). Furthermore, the uncross-linked controls showed a weak signal, indicating that the elevated $^{32}$P signal in the cross-linked samples was due to the formation of cross-links between BPA and phospholipids at those locations. Thus, the phospholipid-binding site observed in our MlaFEDB structure is occupied by phospholipids in vivo.

## Discussion

The results presented here provide mechanistic insights into lipid transport by the Mla system. Our structure reveals that MlaE is structurally related to two bacterial exporters (LPS exporter and MacAB). A role for MlaFEDB in phospholipid export is supported by recent cellular studies in *Acinetobacter baumannii* (*Kamischke et al., 2019*), as well as in vitro experiments with *E. coli* proteins, showing directional lipid transfer from MlaD to MlaC (*Ercan et al., 2019*; *Hughes et al., 2019*). On the other hand, prior studies indicated that the Mla system is an importer (*Malinverni and Silhavy, 2009*; *Chong et al., 2015*; *Powers and Trent, 2018*; *Yeow et al., 2018*), and in vitro reconstitution and transport assays suggest that MlaFEDB may be bi-directional, with a preference for import (retrograde transport) (*Tang, 2020*). Concurrently with our preprint, two other groups also reported structures of the MlaFEDB complex on BioRxiv (*Coudray, 2020*; *Mann, 2020*; *Tang, 2020*), and these structures appear to be similar overall, although the PDB coordinates are not yet available to analyze. While this manuscript was under review, an additional group also published structures of MlaFEDB (*Chi et al., 2020*). Integrating all available data, we present models for both export and

import mediated by MlaFEDB, and the biggest conceptual challenge to each, which remains to be addressed in future work (*Figure 6*).

In anterograde phospholipid export (*Figure 6A*), first, phospholipids must be extracted from the inner membrane and reach the outward-open pocket of MlaE, from either the inner or outer leaflet (*Hughes, 2020*). Due to the structural similarities between MlaE and LptFG, which extracts LPS from the outer leaflet, it is plausible that MlaFEDB may extract PLs from the outer leaflet as well. However, it is also possible that MlaFEDB flips lipids across the bilayer from the inner leaflet to the outward-open pocket observed in our structure before exporting them. Second, substrate reorientation may need to occur such that lipids move through the MlaD pore 'tails-first' to facilitate transfer from MlaD to MlaC, since MlaC binds only the lipid tails (*Huang et al., 2016*; *Ekiert et al., 2017*). This would require a ~ 180° flip of lipid 1 from its orientation bound to MlaE. Thus, the two lipid conformations observed in our structure may be intermediates in the process of lipid flipping, with the extended conformation of lipid 2 representing the halfway-point in the process of completely inverting the orientation of the lipid: between 'tails-down' as if embedded in the outer leaflet of the IM, and 'tails-up' as it traverses the MlaD pore. However, it is also possible that lipids move through the pore in an extended state (resembling lipid 2), perhaps allowing MlaC to first bind to one tail, followed by the second. The lateral channels where the head groups are located may assist in this process, and allow the accommodation of lipids with a variety of head groups. Third, by analogy to the

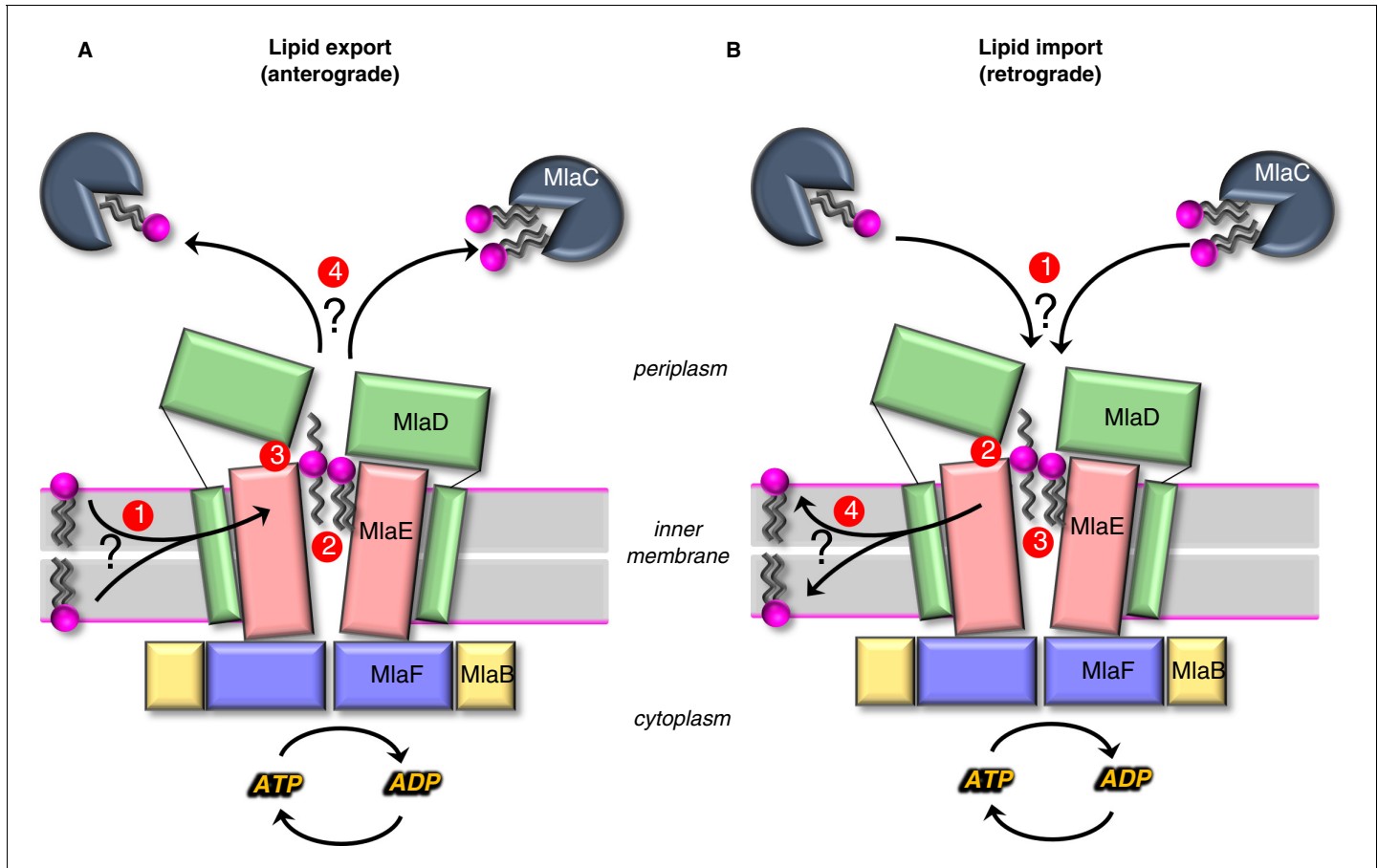

**Figure 6.** Models for lipid transport by MlaFEDB. (**A**) Lipid export model: (1) Lipids are extracted from the IM and transferred to the outward-open pocket by an unknown mechanism. (2) Lipids are reoriented, from 'tails-down' to 'extended' or 'tails-up' configuration. (3) Conformational changes in MlaE coupled to the ATP hydrolysis cycle likely push lipids out of the MlaE pocket and into MlaD pore. (4) Lipids are transferred to MlaC to be shuttled across periplasm to the outer membrane MlaA-OmpC/F complex. MlaC may accommodate one or two phospholipids, or a single larger lipid. (**B**) Lipid import model: (1) Lipids from MlaC are transferred to MlaD, likely dependent on ATP-driven conformational changes in MlaD and MlaE. (2) Lipids travel through the continuous channel from MlaD and are transferred to the outward-open pocket of MlaE. (3) Phospholipids are reoriented 'tails-down', as they are transported between MlaD and MlaE. (4) Lipids are inserted into the inner membrane.

LPS exporter (*Qian et al., 2017*; *Li et al., 2019*; *Owens et al., 2019*), we hypothesize that conformational changes in MlaE may lead to a collapse of the outward-open pocket, extruding the lipids upwards and into the MlaD pore. In the LPS exporter, this step is thought to be linked to ATP binding or hydrolysis, and it is possible that lipid reorientation and pocket collapse occur in a concerted manner, as opposed to sequentially. Fourth, the phospholipid emerges from the MlaD pore 'tails-first' and is transferred to the lipid-binding pocket of an MlaC protein docked on the surface of MlaD. As MlaC proteins from some species are capable of binding multiple phospholipids at one time (*Ekiert et al., 2017*), it is possible that two lipids may be transferred from MlaFEDB to a single MlaC protein, or that multiple MlaC molecules may be involved. To complete the transport process, MlaC would then shuttle phospholipids across the periplasmic space and deliver them to the MlaA-OmpC/F complex for insertion into the OM. Perhaps, the biggest conceptual challenge to models for lipid export revolves around how a tightly bound lipid can be transferred from MlaC to MlaA-OmpC/F in the absence of a direct energy source. It is unclear from available data whether this can occur spontaneously. The structure of the MlaA in complex with OmpF revealed that MlaA lies primarily in the phospholipid-rich inner leaflet of the OM, and is therefore well-positioned to selectively transfer lipids from MlaC directly to inner leaflet (*Abellón-Ruiz et al., 2017*). Such a mechanism is consistent with Mla exporting newly synthesized lipids to the inner leaflet of the OM. However, the H5/H6 'ridge' of MlaA penetrates the outer leaflet of the OM, and molecular dynamics simulations have defined a possible pathway through the MlaA pore to the outer leaflet, in which case anterograde transport by Mla would be counter-intuitive and expected to destabilize the OM (*Abellón-Ruiz et al., 2017*). Future studies will be required to clearly establish the path of lipid translocation through MlaA that connects MlaC to bulk phospholipids in the OM.

In retrograde phospholipid import (*Figure 6B*), first, an interaction between MlaD and MlaC must trigger the transfer of tightly bound lipid(s) from MlaC to MlaD. Second, conformational changes in the outward-open structure of MlaE may drive the transport of lipids from the MlaD pore through the continuous channel into MlaE. Third, based on the conformations of lipid observed, it seems likely that the phospholipids will be reoriented, 'tails-down' into the MlaE lipid-binding site prior to being inserted into the inner membrane. Fourth, lipids must be inserted into the inner membrane. Based on the orientation in our structure, lipids in MlaE would be properly oriented for direct transfer to the outer leaflet of the inner membrane. However, the Mla system was recently reported to catalyze the flipping lipids from the outer leaflet to the inner leaflet (*Hughes, 2020*), raising the possibility that imported lipids may ultimately reach the cytoplasmic leaflet. For the lipid import model, understanding the mechanism of initial lipid transfer from MlaC to MlaD is the biggest conceptual challenge. In vitro, lipid transfer from MlaC to MlaD has not been reported, although the reverse, lipid transfer from MlaD to MlaC, occurs spontaneously (*Huang et al., 2016*; *Ercan et al., 2019*; *Hughes et al., 2019*). MlaC has a very high affinity for lipids (as evidenced by co-purification with lipids [*Ercan et al., 2019*; *Hughes et al., 2019*]), making it likely that ATP hydrolysis is required to drive lipid transfer from MlaC to MlaD. ATP-dependent lipid transfer from MlaC would likely require coupling of ATP-hydrolysis by MlaF in the cytoplasm to the spatially distant MlaD ring. This would require coupled conformational changes transduced via the intervening MlaE subunit, to ultimately produce a conformation in MlaD that is competent to extract lipids from MlaC. Our structure does reveal conformational changes in the MlaD ring relative to the crystal structure (*Ekiert et al., 2017*), and one possibility is that these different conformational states may be related to motions required to facilitate lipid transfer from MlaC to MlaD. In addition, structural work from Chi, et al. revealed movements of the MlaD ring relative to MlaE, wherein the entire ring appears to move away from MlaE as a rigid body.

It is noteworthy that the proposed manner of lipid binding to MlaFEDB differs significantly among all three pre-prints posted around the same time, as well as in the later publication. Our structure and the other *E. coli* MlaFEDB structure described by *Tang, 2020* describe lipid binding to the outward-open pocket of MlaE, while the *Acinetobacter baumannii* MlaFEDB described by *Mann, 2020* describes lipid-binding sites at the pore of MlaD as well as six additional lipid-binding sites in between the pore loops of each of the six MCE domains in the MlaD ring. Tang, et al. observed density assigned to a single phospholipid bound in the outward-open pocket of MlaFEDB, with the head group facing toward the core of the MlaE dimer and tails pointing toward the MlaD pore. This contrasts with our observation of two lipids bound in the outward-open pocket of our structure reconstituted in lipid nanodiscs, where the lipids are bound in roughly the opposite orientation. The

lipid density observed in the MlaE pocket by Chi, et al. resembles the lipid density observed in our map, albeit subjected to twofold averaging about the C2 symmetry axis. The configuration of lipids in their deposited coordinates differ significantly from our model, with both lipids simultaneously adopting the 'extended' conformation we observed. As large portions of the Chi, et al. lipid models are outside the map, while other strong densities in the pocket have been left unmodeled, it is possible that a lipid configuration closer to our model better satisfies the map (*Figure 4—figure supplement 5*). It is also noteworthy that the sample preparation for the first three structures is different. The samples for both the *A. baumannii* and Tang, et al. *E. coli* structures were prepared in detergent. Tang, et al. also added *E. coli* lipid extract to their sample just before grid freezing. In contrast, our structure was reconstituted in lipid nanodiscs. It is unclear if these differences in lipid recognition between the two *E. coli* structures reflect differences in sample preparation, data processing methodology (e.g. asymmetric reconstruction versus application of C2 symmetry), or perhaps represent different snapshots of the transport mechanism, or even differences in lipid conformation when one vs two lipids are bound.

Our data, together with other recent studies, suggest possible mechanisms for phospholipid transport across the cell envelope, and raise the intriguing possibility that Mla may translocate multiple phospholipid substrates each transport cycle, or perhaps accommodate larger lipid substrates like cardiolipin. A defined lipid-binding pocket within MlaE sets the stage for future studies of targeted inhibitors and small molecule modulators of this complex, both in the context of therapeutics against drug-resistant Gram-negative bacteria, and for the study of cell envelope biogenesis in double-membraned bacteria.

# Materials and methods

**Key resources table**

| Reagent type (species) or resource | Designation | Source or reference | Identifiers | Additional information |
|---|---|---|---|---|
| Strain, strain background (*Escherichia coli*) | Rosetta 2 (DE3) | Novagen | #71400 | |
| Strain, strain background (*Escherichia coli*) | *E. coli* K-12 BW25113/wild-type | Coli Genetic Stock Center | N/A | |
| Strain, strain background (*Escherichia coli*) | T7 express *E. coli* | New England Biolabs (NEB) | #C2566H | |
| Genetic reagent (*Escherichia coli*) | *E. coli* K-12 BW25113 Δ*mlaD* | This paper | bBEL182 | Can be obtained from Bhabha-Ekiert lab, NYU Langone. Please see Materials and methods section 'Phenotypic assays for *mla* mutants in *E. coli*' for details of strain construction |
| Genetic reagent (*Escherichia coli*) | *E. coli* K-12 BW25113 Δ*mlaE* | This paper | bBEL183 | Can be obtained from Bhabha-Ekiert lab, NYU Langone. Please see Materials and methods section 'Phenotypic assays for *mla* mutants in *E. coli*' for details of strain construction |
| Antibody | anti-MlaD (rabbit polyclonal) | Other | N/A | (1:10,000). Provided by Henderson lab, University of Queensland |
| Antibody | Anti-Rabbit IgG Antibody IRDye 800CW (Goat polyclonal) | LI-COR Biosciences | #925–32211 RRID:AB_2651127 | (1:10,000) |

*Continued on next page*

*Continued*

| Reagent type (species) or resource | Designation | Source or reference | Identifiers | Additional information |
|---|---|---|---|---|
| Antibody | Penta His Antibody, BSA-free (Mouse monoclonal) | Qiagen | #34660 RRID:AB_2619735 | (1:5000) |
| Antibody | Anti-Mouse IgG Antibody IRDye 800CW (Goat polyclonal) | LI-COR Biosciences | #926–32210 RRID:AB_621842 | (1:10,000) |
| Recombinant DNA reagent | For all plasmid info, please see *Supplementary file 4* | For all plasmid info, please see *Supplementary file 4* | | |
| Commercial assay or kit | Ni Sepharose 6 Fast Flow | GE Healthcare | #17531802 | |
| Commercial assay or kit | Bio-Beads SM-2 Adsorbent Media | Bio-Rad | #1523920 | |
| Commercial assay or kit | Superdex 200 Increase 10/300 | GE Healthcare | #28-9909-44 | |
| Chemical compound, drug | LB Broth, Miller | BD Difco | #244620 | |
| Chemical compound, drug | LB agar pre-mix | BD Difco | #244510 | |
| Chemical compound, drug | Sodium dodecyl sulfate (SDS) | Sigma | #L5750 | |
| Chemical compound, drug | Ethylenediaminetetra acetic acid (EDTA) | Sigma | #ED2SS | |
| Chemical compound, drug | p-benzoyl-l-pheny lalanine (BPA) | Bachem | #F-2800.0005 | |
| Chemical compound, drug | $^{32}$P orthophosphoric acid | PerkinElmer | #NEX053010MC | |
| Chemical compound, drug | *E. coli* polar lipid extract | Avanti | #100600 | |
| Software, algorithm | cryoSPARC v0.6 to 2.12 | *Punjani et al., 2017* | RRID:SCR_016501 | https://cryosparc.com/ |
| Software, algorithm | RELION2.1 to 3.1 | *Kimanius et al., 2016; Fernandez-Leiro and Scheres, 2017* | RRID:SCR_016274 | https://www3.mrc-lmb.cam.ac.uk/relion/index.php/Main_Page |
| Software, algorithm | MotionCor2 | *Zheng et al., 2017* | RRID:SCR_016499 | https://docs.google.com/forms/d/e/1FAIpQLSfAQm5MA81qTx90W9JL6ClzSrM77tytsvyyHh1ZZWrFByhmfQ/viewform |
| Software, algorithm | GCTF | *Zhang, 2016* | RRID:SCR_016500 | https://www.mrc-lmb.cam.ac.uk/kzhang/ |
| Software, algorithm | UCSF Pyem | *Asarnow et al., 2019* | | https://github.com/asarnow/pyem or https://doi.org/10.5281/zenodo.3576630 |
| Software, algorithm | Chimera | *Pettersen et al., 2004* | RRID:SCR_004097 | https://www.cgl.ucsf.edu/chimera/ |
| Software, algorithm | Coot | *Emsley et al., 2010* | RRID:SCR_014222 | https://www2.mrc-lmb.cam.ac.uk/personal/pemsley/coot/ |
| Software, algorithm | Phenix | *Echols et al., 2012; Liebschner et al., 2019* | RRID:SCR_014224 | http://www.phenix-online.org |
| Software, algorithm | MUSCLE | *Edgar, 2004* | RRID:SCR_011812 | |
| Software, algorithm | 3DFSC Processing Server | *Tan et al., 2017* | | https://3dfsc.salk.edu/ |

*Continued*

| Reagent type (species) or resource | Designation | Source or reference | Identifiers | Additional information |
|---|---|---|---|---|
| Software, algorithm | COCOMAPS | *Vangone et al., 2011* | | https://www.molnac.unisa.it/BioTools/cocomaps/ |
| Software, algorithm | CASTp | *Tian et al., 2018* | | http://sts.bioe.uic.edu/castp/index.html?2pk9 |
| Software, algorithm | HOLLOW | *Ho and Gruswitz, 2008* | RRID:SCR_005729 | http://hollow.sourceforge.net/ |
| Software, algorithm | Bendix | *Dahl et al., 2012* | | https://www.ks.uiuc.edu/Research/vmd/plugins/bendix/ |
| Software, algorithm | PyMOL | Schrödinger, LLC | RRID:SCR_000305 | https://pymol.org/2/ |
| Software, algorithm | DALI | *Holm, 2019* | | |

## Expression and purification of MlaFEDB for cryo-EM

To prepare a sample for cryo-EM, plasmid pBEL1200 (*Ekiert et al., 2017*), which contains the *mla-FEDCB* operon with an N-terminal His-tag on MlaD, was transformed into Rosetta 2 (DE3) cells (Novagen). For expression, overnight cultures of Rosetta 2 (DE3)/pBEL1200 were diluted 1:100 in LB (Difco) supplemented with carbenicillin (100 µg/mL) and chloramphenicol (38 µg/mL) and grown at 37˚C with shaking to an OD600 of 0.9, then induced by addition of L-arabinose to a final concentration of 0.2% and continued incubation for 4 hr shaking at 37˚C. Cultures were harvested by centrifugation, and the pellets were resuspended in lysis buffer (50 mM Tris pH 8.0, 300 mM NaCl, 10% glycerol). Cells were lysed by two passes through an Emulsiflex-C3 cell disruptor (Avestin), then centrifuged at 15,000 xg for 30 min to pellet cell debris. The clarified lysates were ultracentrifuged at 37,000 rpm (F37L Fixed-Angle Rotor, Thermo-Fisher) for 45 min to isolate membranes. The membranes were resuspended in membrane solubilization buffer (50 mM Tris pH 8.0, 300 mM NaCl, 10% glycerol, 25 mM DDM) and incubated for 1 hr, rocking at 4˚C. The solubilized membranes were then ultracentrifuged again at 37,000 rpm for 45 min, to pellet any insoluble material. The supernatant was incubated with NiNTA resin (GE Healthcare #17531802) at 4˚C, which was subsequently washed with Ni Wash Buffer (50 mM Tris pH 8.0, 300 mM NaCl, 10 mM imidazole, 10% glycerol, 0.5 mM DDM) and bound proteins eluted with Ni Elution Buffer (50 mM Tris pH 8.0, 300 mM NaCl, 250 mM imidazole, 10% glycerol, 0.5 mM DDM). MlaFEDB containing fractions eluted from the NiNTA column were pooled and concentrated before separation on a Superdex 200 16/60 gel filtration column (GE Healthcare) equilibrated in gel filtration buffer (20 mM Tris pH 8.0, 150 mM NaCl, 0.5 mM DDM). Fractions of MlaFEDB from size exclusion chromatography were pooled and used for incorporation into nanodiscs.

## Reconstitution of MlaFEDB in lipid nanodiscs

For expression of the MSP, MSP1D1, overnight cultures of Rosetta 2 (DE3)/pMSP1D1 (Addgene #20061) were diluted 1:100 in LB (Difco, #244620) supplemented with kanamycin (50 µg/mL) and chloramphenicol (38 µg/mL) and grown at 37˚C with shaking to an OD600 of 0.9, then induced by addition of IPTG to a final concentration of 1 mM and continued incubation for 3 hr shaking at 37˚C. Cultures were harvested by centrifugation, and the pellets were resuspended in lysis buffer (50 mM Tris pH 8.0, 300 mM NaCl, 10 mM imidazole). Cells were lysed by two passes through an Emulsiflex-C3 cell disruptor (Avestin), then centrifuged at 38,000 xg to pellet cell debris. The clarified lysates were incubated with NiNTA resin (GE Healthcare #17531802) at 4˚C, which was subsequently washed with Ni Wash Buffer (50 mM Tris pH 8.0, 300 mM NaCl, 10 mM imidazole) and bound proteins eluted with Ni Elution Buffer (50 mM Tris pH 8.0, 300 mM NaCl, 250 mM imidazole). The His-tag was cleaved using TEV protease.

For nanodisc reconstitution, a protocol was adapted from *Gao et al., 2016*. 2.5 mg of *E. coli* polar lipid extract (Avanti #100600) were dissolved in 1 ml of chloroform in a glass test tube. The chloroform was then evaporated slowly under a stream of argon gas, to produce a thin film of lipids on the bottom of the tube, and further left to dry under vacuum for at least 2 hr. The lipids were

then resuspended in 200 µL of lipid resuspension buffer (20 mM HEPES, 150 mM NaCl, 14 mM DDM, pH 7.4) and sonicated until the mixture was almost clear. The lipids, MSP1D1 and MlaFEDB were mixed at a molar ratio of 400:4:1, respectively, in nanodisc buffer (20 mM HEPES, 150 mM NaCl, pH 7.4), and left to incubate on ice for 30 min. Bio-Beads (Bio-Rad #1523920) were added (20 mg per 1 ml nanodisc mixture) and incubated for 1 hr, rocking at 4°C. A second batch of Bio-Beads were added and incubated at 4°C overnight. The following day, the Bio-Beads were removed and the sample separated on a Superdex 200 16/60 gel filtration column (GE Healthcare) equilibrated in nanodisc buffer (20 mM HEPES, 150 mM NaCl, pH 7.4). Fractions were assessed by SDS-PAGE and negative stain EM, and were pooled and concentrated for cryo-EM grid preparation.

## Cryo-EM grid preparation and data collection

After size exclusion chromatography, 3 µL of MlaFEDB reconstituted into nanodiscs (at a final concentration of 0.95 mg/mL) was applied to 400 mesh quantifoil holey carbon grids 1.2/1.3 glow discharged for 12 s. The sample was then frozen in liquid ethane using the FEI Vitrobot Mark IV. Prescreening of the grids was performed on Talos Arctica TEMs equipped with K2 cameras, operated at 200 kV, and located at PNCC (Portland, OR) or at NYU (New York, NY). Acquisition of the movies used for the final reconstruction was performed on a Titan Krios microscope ('Krios 2') equipped with Gatan K2 Summit camera controlled via Leginon (*Suloway et al., 2005*) and operated at 300 kV (located at the New York Structural Biology Center, NY). Image stacks of 30 frames were collected in super-resolution mode at 0.416 Å per pixel. Data collection parameters are shown in *Supplementary file 1*.

## Cryo-EM data processing

The overall strategy is summarized in *Figure 1—figure supplement 1*. The initial preprocessing steps were all performed within RELION 2.1 (*Kimanius et al., 2016*; *Fernandez-Leiro and Scheres, 2017*). Movies were drift corrected with MotionCor2 (*Zheng et al., 2017*) and CTF estimation was performed using GCTF (*Zhang, 2016*). Approximately 1000 particles were selected manually and subjected to 2D classification. The resulting class averages were used as templates for subsequent automated particle picking of 1,283,606 particles that were extracted with a box size of 300 pixels. The data was then exported in cryoSPARC v 0.6 (*Punjani et al., 2017*) for further processing. After 2D classification, 731,205 particles were used to generate an ab-initio model subjected to heterogenous refinement of three classes. The 3rd class led to a map in agreement with the size and shape of a previously published low resolution reconstruction (*Ekiert et al., 2017*). A second round of heterogeneous classification was then run with the 376,885 particles from this class: only class three led to a high resolution map from 209,224 particles. A curation step was applied to only include particles with assignment probability greater than a threshold of 0.95, reducing the number of particles to 177,513. Particles were then imported back to RELION for additional rounds of local refinement after having re-extracted the particles with a 500 pixel box size. In RELION 3.1-beta, we performed local CTF and aberration refinement and then performed particle polishing (re-doing first motion correction with RELION's own implementation of MotionCorr), which improved the resolution from ~3.5 Å to ~3.3 Å. A second round of CTF and aberration refinement further improved the resolution to ~3.2 Å. The data was then imported into cryoSPARC 2.12 for another round of refinement to 3.05 Å (some default parameters were modified: we used three extra final passes instead of 1, a batchsize epsilon of 0.0005, set the 'optimize per particle defocus' and 'per group ctf parameters' options to true). Average resolution was estimated using gold standard methods and implementations within RELION and cryoSPARC.

Other data processing strategies were explored but failed to bring additional information or improve the resolution: signal subtraction and focused refinement of subdomains (of MlaE, or MlaFEB with and without C2 symmetry), other rounds of 3D classification, or further restricting the selection of particles to the best ones relying on the probability distributions computed in RELION (rlnLogLikeliContribution rlnMaxValueProbabilityDistribution).

Transfer of data from cryoSPARC to RELION was performed using UCSF Pyem (*Asarnow et al., 2019*).

## Model building

The following models were used as a starting models for the MlaFEDB structure: for MlaFB, the X-ray structure, PDB ID: 6XGY (*Kolich et al., 2020*); for the MCE domain protein MlaD, PDB ID: 5UW2 (*Ekiert et al., 2017*); and for MlaE, a computationally predicted model (*Ovchinnikov et al., 2015*; *Ekiert et al., 2017*). Domains were docked as rigid bodies in Chimera (*Pettersen et al., 2004*), and manual model building was done in COOT (*Emsley et al., 2010*). The models were then iteratively refined using the real_space_refine algorithm in PHENIX (*Echols et al., 2012*; *Liebschner et al., 2019*), with rounds of manual model building in between using COOT. The six TM helices from the six MlaD subunits, not present in the construct used of the X-ray structure, were built de novo, but the loops connecting those helices to the core MCE domains were too flexible to be modeled. For the two MlaD TM$^{A/D}$ helices contacting IF2 and TM3 of MlaE, as well as the two MlaD TM$^{B/E}$ helices that contact IF1 on MlaE, the clear side chain density allowed unambiguous assignment of the sequence register. For the final MlaD TM$^{C/F}$ helices contacting IF1 of MlaE near the N-Terminus, the EM map was filtered to 6 Å to better visualize the density, as these helices are more flexible. The close helical packing geometry between IF1 and MlaD-TM$^{B/E}$ enforces a strong preference for glycine residues at the positions of closest contact (Gly21$^{MlaE}$ and Gly11$^{MlaD}$, ~3.4 Å Cα-Cα) (*Figure 3—figure supplement 1A*). Gly is present at MlaD position 11 in 13/13 sequences analyzed, and at MlaE position 21 in 12/13 sequences analyzed (*Figure 3—figure supplement 1B, C*), suggesting that the interactions between IF1 and MlaD-TM$^{B/E}$ are specific and conserved. Gly11 of MlaD-TM$^{B/E}$ and residues of IF1 are part of a larger interaction motif (17-LxxFGxxxL-25) (*Figure 3—figure supplement 1A*). MlaD-TM$^{C/F}$ appears to interact with IF1 in a similar manner in the vicinity of MlaE Gly10 (Gly10$^{MlaE}$ and Gly11$^{MlaD}$, ~3.5 Å Cα-Cα), which is part of a very similar conserved motif (6-LxxLGxxxI-14; *Figure 3—figure supplement 1*). While density for side chains in MlaD TM$^{C/F}$ is weak, the similarity in helix packing geometry and these two binding sites, along with only one available Gly for close helix packing in the MlaD TM helix suggest that the same surface of the MlaD TM is used for IF1 binding in these chains C and F as well. Consequently, we have used this Gly-Gly close packing to establish a likely sequence register for these TM helices. Due to the lower resolution, we did not model the side chains of these residues explicitly.

The MlaE region displayed the highest local resolution (below 3 Å in its core) and was almost entirely modeled. The two most flexible regions were the extremities of the interfacial helices IF1 (the N-Terminus and the connection to the IF2/TM1 helix). Due to a lack of density on the N-terminus of IF1, residues 1–4 could not be modeled. Within the MlaE dimer and at the interface with MlaD, we identified two clearly defined densities that corresponded to the shape and size of phospholipids, which are present in our reconstitution. Two phosphatidylethanolamine molecules (code: PEF) were manually placed into the densities. The phosphate, glycerol backbone, and most of the C16 fatty acid chains could readily be placed in the map, but the ethanolamine portion of the head group was removed due to a lack of density, meaning this part may be flexible and/or non-specific to a certain type of lipid. In reality, our MlaFEDB sample likely contained a heterogeneous mixture of lipids bound, with a range of head groups and acyl chain lengths/unsaturations. Although the resulting ligands resemble phosphatidic acid, we retain the PEF/phosphatidylethanolamine designation, as PE is the most abundant PL species in *E. coli* but phosphatidic acid is relatively scarce.

We have modeled two PE molecules bound to the transporter simultaneously, as this best explains all the available information, including: (1) The two densities are well-resolved and do not cross each other; (2) after refinement, the atoms of the lipids are roughly within van der Waals distance of each other and nearby protein atoms, without excessive clashes and in line with expectations for flexible/heterogeneous ligands at this resolution; (3) while the tight packing of two lipids fills the MlaE pocket, binding of a single lipid would leave the pocket ~50% empty; based upon the observed protein-lipid interactions, it is difficult to envision how single lipid molecules could be bound in the pocket yet be constraint of the observed conformation of lipid 1 and lipid 2 unless a second lipid is simultaneously present; (4) we performed various focused 3D classification with variable masks and regularization parameters in RELION, as well as 3D variability analysis in cluster mode in cryoSPARC with and without a mask. While the resulting maps were generally of lower quality, the reconstructions containing clear lipid-like densities most closely resembled the configuration of the two lipids modeled in our structure.

Using both the high-resolution map and its filtered version at 6 Å, we also modeled both coils of the MSP belt surrounding the edges of the nanodisc (starting with the ones modeled in PDB: 6CM1). These MSP belts took the form of two relatively featureless tubes of density. Consequently, their position is modeled as using polyalanine helices, and we were able to account for ~160 of the expected ~190 residues for MSP1D1.

The final model of the MlaFEDB complex is nearly complete, with two noteworthy areas of disorder. First, the loops between the TM helices and the MCE domains of MlaD could not be resolved (5–8 residues disordered in each). Second, the last 32 residues at the C-terminus of each MlaD chain (residues 153–183), which were disordered in previous X-ray structures (*Ekiert et al., 2017*), were also not visible in our EM structure.

## Structure analysis and bioinformatics

As structural deviations between MlaE and other ABC TMDs made database searches more difficult, we conducted a Dali search (*Holm, 2019*) initiated with MlaF to recover all of the PDB structures containing an ABC domain. These structures were then manually curated based upon the presence or absence of TMDs and further classified based upon the TMD fold.

In order to assess the conservation of MlaD and MlaE sequences across species, we identified at least one 'representative' species from each major bacterial order, across the entire bacterial kingdom. Within each order, we selected 'representative' species, which was typically one of the most widely studied members and/or a special most impacting human health (prior to an examination of the sequences, to avoid bias). For each representative species, we searched the reference genome using BLAST to identify possible homologs of *E. coli* MlaE, MlaD, MlaC, and MlaA. We did not search directly for MlaF or MlaB homologs, as ABC ATPases and STAS domain proteins unrelated to Mla are common in bacteria. Of 65 species analyzed, only 13 were determined to encode what appeared to be functional MCE transporters that were 'true homologs' of Mla (*Supplementary file 3*). To be included in this group, the species must encode a homolog of MlaD (single MCE domain without a long C-terminal helical region (less than ~50 residues) and also homologs of MlaE, MlaC, and MlaA in its genome). In every case, MlaE, MlaD, and MlaC were encoded just downstream of an MlaF-like ABC subunit, and just upstream of an MlaB-like protein (except in *Rickettsia rickettsii*, which appears to lack MlaB). Sometimes MlaA was encoded in the same operon, while sometimes it was encoded elsewhere in the genome. The resulting 'True Mla' homologs were subsequently used for sequence analysis. Sequence alignments were generated using MUSCLE (*Edgar, 2004*) and visualized using JalView (*Waterhouse et al., 2009*).

The 3DFSC in *Supplementary file 2* was measured using the Remote 3DFSC Processing Server (*Tan et al., 2017*). The interfaces between the different subunits of MlaFEDB, Lpt and ABCA/G proteins were analyzed using the COCOMAPS server (*Vangone et al., 2011*). The area of the cavities of MlaE and LptFG were estimated using CASTp (*Tian et al., 2018*) and HOLLOW (*Ho and Gruswitz, 2008*). The curvature of the IF2-TM1 helices was analyzed using Bendix (*Dahl et al., 2012*), and the corresponding figures generated with VMD software support which is developed with NIH support by the Theoretical and Computational Biophysics group at the Beckman Institute, University of Illinois at Urbana-Champaign. All other figures were made with Chimera (*Pettersen et al., 2004*) or PyMOL (Schrödinger, LLC). The PyMOL plugin, anglebetweenhelices (Schrödinger, LLC), was used to compute the angle between IF1 of MlaE and the TM helices of MlaD.

## Phenotypic assays for *mla* mutants in *E. coli*

Knockouts of *mlaD* and *mlaE* were constructed in *E. coli* BW25113 by P1 transduction from the Keio collection (*Baba et al., 2006*), followed by excision of the antibiotic resistance cassettes using pCP20 (*Cherepanov and Wackernagel, 1995*). Serial dilutions of the strains in 96-well plates were manually spotted (2 µL each) on plates containing LB agar or LB agar supplemented with 0.2% SDS and 0.30–0.35 mM EDTA, and incubated for 16 hr at 37°C. We find that this growth assay is very sensitive to the reagents used, particularly the LB agar (see *Kolich et al., 2020*). For the experiments reported here, we used Difco LB agar pre-mix (BD Difco #244510), a 10% stock solution of SDS (Sigma L5750), and a 500 mM stock solution of EDTA, pH 8.0 (Sigma ED2SS). Furthermore, we note that when the agar plates were incubated longer than 16 hr, we began to observe some clearing/loss of pigmentation of the bacterial spot dilutions.

For complementation and/or testing the functionality of the various MlaD and MlaE mutants, a pBAD-derived plasmid harboring the *mlaFEDCB* operon was transformed into the appropriate knockout strain. To test the functionality of mutations in MlaD, we transformed the *mlaD* knockout strain with pBEL1198 (*mlaFEDCB* operon N-terminal His-tag on MlaE, see *Supplementary file 4*), or derivatives of this plasmid harboring the desired mutations in MlaD (MlaD TM replaced with LptC TM (pBEL2139), MlaD TM replaced with LetB TM (pBEL2138)), Gly11Phe (pBEL2290), Ile12Ala (pBEL2291), Gly11Phe and Ile12Ala (pBEL2292), Phe13Ala (pBEL2294), Ala17Phe (pBEL2295), Ala20-Phe (pBEL2296), Val24Phe (pBEL2297), and a triple mutant of Ala17Phe, Ala20Phe and Val24Phe (pBEL2298), see (*Supplementary file 4*). For the MlaE mutants, we transformed the *mlaE* knockout strain with pBEL1200 (*mlaFEDCB* operon with N-terminal His-tag on MlaD, see *Supplementary file 4*), or derivatives of this plasmid harboring the desired mutations in MlaE (IF1 1–15 aa deletion (pBEL2093), IF1 1–25 aa deletion (pBEL2132), IF1 1–30 aa deletion (pBEL2092), IF1 1–39 aa deletion (pBEL2133), Tyr81Ala (pBEL2099), Tyr81Trp (pBEL2100), Arg97Ala (pBEL2098), Glu98Ala (pBEL2094), Lys205Ala (pBEL2095), and Asp250Ala (pBEL2097), see *Supplementary file 4*). We found that leaky expression from the pBAD promoter was sufficient for complementation of the phenotypes of both the *mlaD* and *mlaE* knockout strains, and thus no L-arabinose was added. We suspect that these constructs significantly over-produce MlaFEDCB proteins, and while some mutants tested confirmed our ability to detect loss-of-function mutations, it is possible that this over-expression may mask the impact of mutations that cause a moderate reduction in MlaFEDB activity.

## Expression and purification of MlaFEDB mutants

MlaFEDB mutants were expressed and purified by NiNTA affinity chromatography as described above. For studies involving mutations in MlaD, we used a construct with the WT *mlaFEDCB* operon with an N-terminal His-tag on MlaE (pBEL1198), or derivatives in which the MlaD TM was replaced with LptC TM (pBEL2139) or LetB TM (pBEL2138), or a triple mutation of MlaD Ala17Phe, Ala20Phe and Val24Phe (pBEL2298) was introduced. For studies involving mutations in MlaE, we used a construct with the WT *mlaFEDCB* operon with an N-terminal His-tag on MlaD (pBEL1200), or derivatives with the MlaE IF1 1–15 aa deletion (pBEL2093), MlaE IF1 1–25 aa deletion (pBEL2132), MlaE IF1 1–30 aa deletion (pBEL2092), or MlaE IF1 1–39 aa deletion (pBEL2133).

## Western blot to detect MlaD TM chimera mutants

In order to assess the expression and cellular localization of the MlaD mutants with the native TM replaced by the TM from LetB or LptC, each of the pBEL1198 derived plasmids (WT operon [pBEL1198], MlaD TM replaced with LptC TM [pBEL2139] and MlaD TM replaced with LetB TM [pBEL2138], see *Supplementary file 4*) were expressed and purified as described above (see Expression and purification of MlaFEDB). Following cell lysis and a low-speed spin to remove cell debris, a sample was collected, which we refer to as the 'whole cell lysate'. The membranes were then isolated and solubilized as described above, and a sample was taken from the solubilized membranes, which we refer to as the 'membrane fraction'. Of each sample, 10 µL were separated on an SDS-PAGE gel and transferred to a nitrocellulose membrane. The membranes were blocked in PBST containing 5% milk for 1 hr. The membranes were then incubated with primary antibody (rabbit polyclonal anti-MlaD (provided by Henderson lab, University of Queensland) at a dilution of 1:10,000) in PBST + 5% milk overnight at 4°C. The membranes were then washed three times with PBST and were incubated with goat anti-rabbit IgG polyclonal antibody (IRDye 800CW, LI-COR Biosciences #925–32211 at a dilution of 1:10,000) in PBST + 5% milk for 1 hr. The membranes were then washed three times with PBST and imaged using a LI-COR (LI-COR Biosciences).

## Lipid cross-linking experiments

This method was adapted from *Isom et al., 2020*. T7express *E. coli* (NEB) were co-transformed with (1) plasmids to express MlaFEDCB (either the WT proteins using pBEL1198, or derivatives of this plasmid expressing Amber mutant MlaE variants for BPA incorporation (Tyr81Bpa [pBEL2057], Val77Bpa [pBEL2060], Leu78Bpa [pBEL2061], Leu70Bpa [pBEL2062], Leu99Bpa [pBEL2063], Trp149Bpa [pBEL2065] or Phe209Bpa [pBEL2066]); and (2) pEVOL-pBpF (Addgene #31190), which encodes a tRNA synthetase/tRNA pair for the in vivo incorporation p-benzoyl-l-phenylalanine (BPA) in *E. coli* proteins at the amber stop codon, TAG (*Chin et al., 2002*; *Isom et al., 2017*). Bacterial

colonies were inoculated in LB broth supplemented with carbenicillin (100 µg/mL), chloramphenicol (38 µg/mL) and 1% glucose, and grown overnight at 37˚C. The following day, bacteria were pelleted and resuspended in $^{32}$P Labeling Medium (a low phosphate minimal media: 1 mM Na2HP04, 1 mM KH2PO4, 50 mM NH4Cl, 5 mM Na2SO4, 2 mM MgSO4, 20 mM Na2-Succinate, 0.2x trace metals and 0.2% glucose) supplemented with carbenicillin (100 µg/mL) and chloramphenicol (38 µg/mL) and inoculated 1:33 in the 10 mL of the same medium. Bacteria were grown to OD 1.0 and a final concentration of 0.2% L-arabinose and 0.5 mM BPA (Bachem, #F-2800.0005), alongside 375 µCi $^{32}$P orthophosphoric acid (PerkinElmer, #NEX053010MC) were added and left to induce overnight.

The following day, the cultures were spun down and resuspended in 1 mL of PBS, and the 'cross-linked' samples underwent cross-linking by treatment with 365 nM UV in a Spectrolinker for 30 min. Both the cross-linked and uncross-linked cells were then spun down and resuspended in 1 mL of lysozyme-EDTA resuspension buffer (50 mM Tris pH 8.0, 300 mM NaCl, 10 mM imidazole, 1 mg/mL lysozyme, 0.5 mM EDTA, 25 U/mL benzonase) and were incubated for 1 hr at room temperature. The cells then underwent eight cycles of freeze-thaw lysis by alternating between liquid nitrogen and a 37˚C heat block. The lysate was pelleted at 20,000 xg for 15 min, and the pellets were resuspended in 133 µL of membrane resuspension buffer (50 mM Tris pH 8.0, 300 mM NaCl, 10% glycerol and 25 mM DDM), and incubated, shaking, for 1 hr. The sample volume was then increased to 1 mL using 10 mM wash buffer (50 mM Tris pH 8.0, 300 mM NaCl, 10 mM imidazole) and insoluble material was pelleted at 20,000 xg for 15 min. Each supernatant was then mixed with 25 µL of nickel beads (Ni Sepharose 6 Fast Flow) for 30 min. The beads were pelleted at 500 xg for 1 min and the supernatant collected. The beads were then washed four times with 40 mM wash buffer (50 mM Tris pH 8.0, 300 mM NaCl, 40 mM imidazole, 10% glycerol, 0.5 mM DDM) and finally resuspended in 50 µL of elution buffer (50 mM Tris pH 8.0, 300 mM NaCl, 300 mM imidazole, 10% glycerol, 0.5 mM DDM). The samples were then mixed with 5x SDS-PAGE loading buffer, and the beads spun down at 12,000 xg for 2 min. Eluted protein was analyzed by SDS-PAGE and stained using InstantBlue Protein Stain (Expedeon, #isb1l). Relative loading of the MlaE monomer band on the gel was estimated integrating the density of the corresponding bands in the InstantBlue-stained gel in ImageJ (*Rueden et al., 2017*), and this was used to normalize the amount of MlaE monomer loaded on a second gel, to enable more accurate comparisons between samples. The normalized gel was stained with InstantBlue and $^{32}$P signal was detected using a phosphor screen and scanned on a Typhoon scanner (Amersham). Three replicates of the experiment were performed, starting with protein expression. NB: earlier protocols using urea solubilization (*Coudray, 2020*) gave globally similar results but with variation in cross-linking efficiency between biological replicates; the improved protocol used here, purifying MlaFEDB under native conditions (without urea), has much lower variation between replicates.

## Western blots to detect MlaE in cross-linked samples

Samples were grown and induced as described above (see Lipid cross-linking experiments), but in the absence of $^{32}$P orthophosphoric acid. The Western blot was done as described above (in Western blots to detect MlaD TM mutants), but with an anti-His antibody (Qiagen, #34660 at a dilution of 1:5000) as primary, to detect His-tagged MlaE, and goat anti-mouse IgG polyclonal antibody (IRDye 800CW, LI-COR Biosciences #926–32210 at a dilution of 1:10,000) as the secondary antibody.

## Acknowledgements

We thank Marisa Lopez-Redondo (NYU) and Yuan Gao (UCSF) for guidance with nanodisc reconstitution, and members of the Bhabha/Ekiert labs for helpful discussions. We thank Ian Henderson (University of Queensland) for providing anti-MlaD antibodies. We thank Noelle Antao, Juliana Ilmain, Dhenesh Puvanendren, Rachel Redler and Casey Vieni for critical reading and feedback on our manuscript. We gratefully acknowledge the following funding sources: NIH grant R35GM128777 (DCE), NIH grant R00GM112982, Damon Runyon Cancer Research Foundation grant DFS-20-16 and Pew Charitable Trusts PEW-00033055 (GB). American Heart Association postdoctoral fellowship 20POST35210202 (GLI), NIH T32 predoctoral training grant T32 GM088118 (MRM). Pre-screening for cryo-EM sample optimization was carried out at the NYU cryo-EM core facility and the Pacific Northwest Center for Cryo-EM. Cryo-EM data were collected at the Simons Electron Microscopy Center and National Resource for Automated Molecular Microscopy located at the New York

Structural Biology Center, supported by grants from the Simons Foundation (SF349247), NYSTAR, and the NIH National Institute of General Medical Sciences (GM103310) with additional support from Agouron Institute (F00316) and NIH (OD019994). A portion of this research was supported by NIH grant U24GM129547 and performed at the Pacific Northwest Center for Cryo-EM at Oregon Health and Sciences University (OHSU) and accessed through EMSL (grid.436923.9), a DOE Office of Science User Facility sponsored by the Office of Biological and Environmental Research. EM data processing has utilized computing resources at the HPC Facility at NYU, and we thank the HPC team. Molecular graphics and analyses performed with UCSF Chimera, developed by the Resource for Biocomputing, Visualization, and Informatics at the University of California, San Francisco, with support from NIH P41-GM103311.

## Additional information

### Funding

| Funder | Grant reference number | Author |
| --- | --- | --- |
| National Institutes of Health | R35GM128777 | Damian C Ekiert |
| National Institutes of Health | R00GM112982 | Gira Bhabha |
| Damon Runyon Cancer Research Foundation | DFS-20-16 | Gira Bhabha |
| Pew Charitable Trusts | PEW-00033055 | Gira Bhabha |
| American Heart Association | 20POST35210202 | Georgia L Isom |
| National Institutes of Health | T32 GM088118 | Mark R MacRae |

The funders had no role in study design, data collection and interpretation, or the decision to submit the work for publication.

### Author contributions

Nicolas Coudray, Mark R MacRae, Formal analysis, Validation, Investigation, Visualization, Writing - original draft, Writing - review and editing; Georgia L Isom, Formal analysis, Funding acquisition, Validation, Investigation, Visualization, Writing - original draft, Writing - review and editing; Mariyah N Saiduddin, Formal analysis, Validation, Writing - review and editing; Gira Bhabha, Damian C Ekiert, Conceptualization, Supervision, Funding acquisition, Validation, Investigation, Writing - original draft, Writing - review and editing

### Author ORCIDs

Nicolas Coudray https://orcid.org/0000-0002-6050-2219
Georgia L Isom https://orcid.org/0000-0001-9023-705X
Mark R MacRae https://orcid.org/0000-0003-4941-9526
Gira Bhabha http://orcid.org/0000-0003-0624-6178
Damian C Ekiert https://orcid.org/0000-0002-2570-0404

### Decision letter and Author response

Decision letter https://doi.org/10.7554/eLife.62518.sa1
Author response https://doi.org/10.7554/eLife.62518.sa2

## Additional files

### Supplementary files

- Supplementary file 1. Data collection parameters for cryo-EM structure of MlaFEDB.
- Supplementary file 2. Data refinement statistics for cryo-EM structure of MlaFEDB.
- Supplementary file 3. Species included in analysis of Mla sequence conservation.
- Supplementary file 4. Plasmids used in this study.

- Transparent reporting form

### Data availability

The model has been deposited in PDB under the accession code 6XBD and the map has been deposited in EMDB under the accession code EMD-22116. Raw data were deposited into EMPIAR (EMPIAR-10536). Plasmids generated in this study will be deposited in Addgene.

The following datasets were generated:

| Author(s) | Year | Dataset title | Dataset URL | Database and Identifier |
|---|---|---|---|---|
| Coudray N, Isom GL, MacRae MR, Saiduddin M, Ekiert DC, Bhabha G | 2020 | Cryo-EM structure of MlaFEDB in nanodiscs with phospholipid substrates | https://www.rcsb.org/structure/6XBD | RCSB Protein Data Bank, 6XBD |
| Coudray N, Isom GL, MacRae MR, Saiduddin M, Ekiert DC, Bhabha G | 2020 | Cryo-EM structure of MlaFEDB in nanodiscs with phospholipid substrates | https://www.ebi.ac.uk/pdbe/entry/emdb/EMD-22116 | Electron Microscopy Data Bank, EMD-22116 |
| Coudray N, Isom GL, MacRae MR, Saiduddin MN, Ekiert DC, Bhabha G | 2020 | Single particle cryo-EM dataset for MlaFEDB from E. coli in nanodisc | https://www.ebi.ac.uk/pdbe/emdb/empiar/entry/10536/ | Electron Microscopy Public Image Archive, EMPIAR-10536 |

The following previously published datasets were used:

| Author(s) | Year | Dataset title | Dataset URL | Database and Identifier |
|---|---|---|---|---|
| Chang Y, Bhabha G, Ekiert DC | 2020 | Crystal structure of E. coli MlaFB ABC transport subunits in the dimeric state | https://www.rcsb.org/structure/6XGY | RCSB Protein Data Bank, 6XGY |
| Bhabha G, Ekiert DC | 2017 | Structure of E. coli MCE protein MlaD, periplasmic domain | https://www.rcsb.org/structure/5UW2 | RCSB Protein Data Bank, 5UW2 |

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
