## [Decision Letter]

**Acceptance summary:**

The study by Coudray et al. describes a cryo-EM structure of the ABC transporter MlaFEDB from *E. coli* after being reconstituted into nanodiscs. The structural analysis reveals how the MlaEF ABC transporter interacts with MlaD on the periplasmic and MlaB on the cytosolic side of the membrane. The authors also report structure-function mutants and assembly studies to gain insights into certain features, such as the MlaD membrane helices that interact with MlaE and specific interactions with lipids that likely relate to a transport pathway. in vivo data confirm crosslinking between residues lining the pocket and these lipids, which seem to be in the process of being exported or imported. Together with studies by other groups, the value of this structure, new loss-of-function mutants, and other observations is that they will help elucidate the still enigmatic way in which the Mla system functions to transport lipids.

**Decision letter after peer review:**

Thank you for submitting your article "Structure of bacterial phospholipid transporter MlaFEDB with substrate bound" for consideration by *eLife*. Your article has been reviewed by two peer reviewers, and the evaluation has been overseen by a Reviewing Editor and Olga Boudker as the Senior Editor. The following individuals involved in the review of your submission have agreed to reveal their identity: Bert van den Berg (Reviewer #1); Jochen Zimmer (Reviewer #2).

The reviewers have discussed the reviews with one another and the Reviewing Editor has drafted this decision to help you prepare a revised submission.

Summary:

The paper by Ekiert and colleagues describes a cryo-EM structure of the ABC transporter MlaFEDB from *E. coli* and reconstituted into nanodiscs. The structures reveal how the MlaEF ABC transporter interacts with MlaD on the periplasmic and MlaB on the cytosolic side of the membrane. Overall, the structure is fascinating and gives an excellent idea of the overall conformation of the complex and the orientations of the various subunits relative to each other and the membranes. The authors also report structure-function studies using an SDS-EDTA sensitivity assay to gain insights into several structural features, such as the MlaD membrane helices that interact with MlaE and specific interactions with lipids that are presumed to relate to a transport pathway. One of the most interesting observations in the structure is the presence of the two bound phospholipid molecules in an outward open binding pocket in the MlaE membrane subunit. in vivo data confirm crosslinking between residues lining the pocket and PL. The phospholipids have very different conformations and may be in the process of being exported (or imported). The manuscript is well written and the figures are informative. Together with other papers from additional groups that report on the structure of the same complex, the value of the structure and other observations made in this paper is that they will help elucidating the still enigmatic way in which the Mla system functions, particularly with respect to the directionality of lipid transport.

Essential revisions:

1) Figure 3, please augment the analysis of MlaD chimeras used to test the significance and determinants of the N-terminal TM helix interactions. Instead of replacing the entire helix, please test interactions highlighted in Figure 3—figure supplement 1 by site directed mutagenesis, focused on those involved in the LxxF/LG motifs. This is especially important considering the modest resolution in this region and the challenges associated with assigning the register of the helix.

2) Expand the Discussion and bibliography to acknowledge the presence of published, competitive studies. Please comment on shared discoveries versus the remaining functional questions that require further work.

3) The Discussion should also include the published structure of the MlaA-OmpF complex. Do the authors agree that the MlaA structure suggests that any PL received from MlaC during anterograde transport would be deposited in the OM outer leaflet, which would be counterproductive? Does this rule out an anterograde transport mechanism?

4) The authors wrote: "Some of the differences between these structures could be ascribed to differences between the *E. coli* and *A. baumannii* transporters [sp?]". This is vague, either be more specific or remove?

5) Abstract: Please explain 'MCE ring'.

6) Reference Kolich, L. et al., Please update to accepted manuscript.

7) Please explain: "…that cardiolipin bound to MlaE would have to adopt a somewhat different configuration.….., in order to covalently link the two phosphate groups closer together." What does "covalently link" refer to?

8) Figure 2B: Fix the Spelling of *A. baumannii.*

---

## [Author Response]

Essential revisions:1) Figure 3, please augment the analysis of MlaD chimeras used to test the significance and determinants of the N-terminal TM helix interactions. Instead of replacing the entire helix, please test interactions highlighted in Figure 3—figure supplement 1 by site directed mutagenesis, focused on those involved in the LxxF/LG motifs. This is especially important considering the modest resolution in this region and the challenges associated with assigning the register of the helix.

We have now tested several additional point mutations in the MlaD TM to better understand the importance of the MlaE interactions we observed in the structure. In addition to contacts in the vicinity of the LxxF/LG motifs specifically mentioned by the reviewers, we have also made mutations at other sites of contact between the MlaD TMs and MlaE. The impact of mutations in the vicinity of the LxxF/LG motifs is consistent with our previous data for a deletion of MlaE residues 1-25 that removes both LxxF/LG motifs. Both results indicate that contacts between the IF1 region and 4/6 MlaD TMs are not strictly required for MlaFEDB assembly (assessed by pull-down) or function (in our cell based assay). But intriguingly, our new mutants show that a triple mutant designed to disrupt the final 2/6 MlaD TM interfaces with MlaE abolishes function in cells. Probing this mutant further, we find that, surprisingly, it is still capable of forming a complex. We present these new data in Figure 3 and Figure 3—figure supplement 2 and discuss these results in the subsection “Interactions between MlaE and MlaD”. We think the triple mutant may open up an interesting new research direction that we will pursue in the future, as several mechanistic hypotheses can be generated from the new data. We thank the reviewers for making the suggestion to further test the specificity of the MlaED interaction.

2) Expand the Discussion and bibliography to acknowledge the presence of published, competitive studies. Please comment on shared discoveries versus the remaining functional questions that require further work.

In the submitted manuscript, we had already discussed the two other pre-prints that were posted around the same time as our pre-print. We now also briefly comment on the work from Chi, et al., which was available only after our manuscript was in review, in the Discussion.

3) The Discussion should also include the published structure of the MlaA-OmpF complex. Do the authors agree that the MlaA structure suggests that any PL received from MlaC during anterograde transport would be deposited in the OM outer leaflet, which would be counterproductive? Does this rule out an anterograde transport mechanism?

Our current work focuses only on the inner membrane MlaFEDB complex and does not shed any new light on the MlaA-OmpF complex in the outer membrane, or a direct assessment of the directionality of transport. However, we have now added a brief discussion of previous work on the MlaA-OmpF structure and its implications in the Discussion. We agree that the proposed mechanism for lipid transfer through the MlaA pore is inconsistent with anterograde transport, and would indeed seem to be counterproductive. However, in our opinion, the structure of MlaA could plausibly allow for lipid transfer to/from the inner leaflet instead; indeed, as MlaA resides primarily within the inner leaflet, it is arguably better positioned to selectively deliver lipids to the inner leaflet instead of the outer leaflet. If MlaA created a pathway to the inner leaflet instead, anterograde transport would make sense. The molecular dynamics simulations reported along with the structure provide a possible route for lipid transfer to/from the outer leaflet, but we believe that there still remains significant ambiguity in the direction of transport by Mla and believe that presenting both models along with their challenges may be helpful for readers and the field.

4) The authors wrote: "Some of the differences between these structures could be ascribed to differences between the E. coli and A. baumannii transporters [sp?]". This is vague, either be more specific or remove?

We agree and have removed this sentence.

5) Abstract: Please explain 'MCE ring'.

We have reworded this sentence to clarify the meaning

6) Reference Kolich, L. et al., Please update to accepted manuscript.

The reference was updated.

7) Please explain: "…that cardiolipin bound to MlaE would have to adopt a somewhat different configuration.….., in order to covalently link the two phosphate groups closer together." What does "covalently link" refer to?

This sentence was unclear and we have revised it to be more specific.

8) Figure 2B: Fix the Spelling of A. baumannii

The spelling was corrected.